# WebAggregator: Scaling Complex Logical Information Aggregation for Web Agent Foundation Models

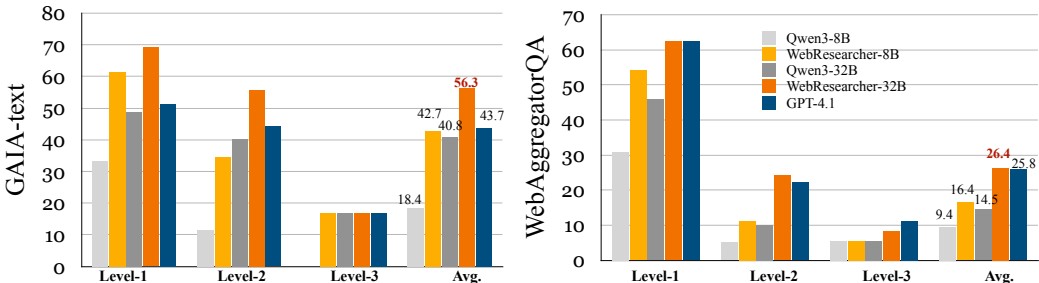

Figure 1: The Pass@1 performance of our WebAggregator models, tuned on the automatically constructed training resource, WebAggregatorQA, is comparable to or even exceeds that of GPT-4.1 on both GAIA-text and the more challenging WebAggregatorQA test set.

## Abstract

Deep research web agents must not only retrieve information from diverse sources such as web environments, files, and multimodal inputs, but more importantly, they need to rigorously analyze and aggregate knowledge in order to generate high-quality, insightful research. However, existing open-source deep research agent systems predominantly focus on enhancing *information seeking* capabilities of web agents to *locate* specific information, while overlooking the essential need for *information aggregation*, which would limit their ability to generate coherent insights or support in-depth research.

In this paper, we propose a paradigm for scalably constructing verifiable training datasets for web agents, by framing data construction as an agentic task grounded in real web pages while placing additional focus on developing fine-grained rules that enable complex information aggregation. Our approach synthesizes tasks by first collecting information through *proactive online web exploring* on the real web environment, followed by *Complex Aggregation Logic Injection* to compose the verifiable question-answer pairs from aggregated knowledge snippets, covering over 12 logical operations. The resulting dataset contains about 10K samples across 50K websites, covering more than 11 domains. Based on an open-source agent framework, SmolAgents, we collect supervised fine-tuning trajectories to develop a series of foundation models, named WebAggregator. WebAggregator-8B matches the performance of GPT-4.1, while the 32B variant surpasses GPT-4.1 by more than 10% on GAIA-text and closely approaches the performance of Claude-3.7-sonnet.

Moreover, given the limited availability of benchmarks that evaluate web agents' information aggregation abilities, we construct a human-annotated evaluation split of WebAggregatorQA as a challenging test set. On this benchmark, Claude-3.7-sonnet only achieves 28%, and GPT-4.1 scores 25.8%, and even after retrieving all of the references, they still struggle on WebAggregatorQA, highlighting the need to strengthen the information aggregation capabilities of web agent foundations.

# 1 INTRODUCTION

DeepResearch agent systems (OpenAI, 2025; Monica.Im, 2025) are built upon foundational large language models (LLMs), aiming to perform complex, human-level tasks. Achieving this level of capability requires not only effective *information seeking*, using web-interactive tools to retrieve accurate and relevant knowledge, but more importantly, *information aggregation*, where retrieved materials are synthesized into coherent, novel insights in the spirit of expert human authors (Bereiter & Scardamalia, 1987; Flower & Hayes, 1981).

Developing web agents with human-level task composition capabilities fundamentally requires large-scale training corpora that explicitly capture both *information seeking* and *aggregation* behaviors. Yet, such datasets remain scarce. Existing multi-hop QA datasets (Yang et al., 2018; Talmor et al., 2021; Trivedi et al., 2022) rarely involve authentic web interactions, and can often be solved from the models' parametric knowledge alone. More recent web agent datasets (Shi et al., 2025; Wu et al., 2025a; Tao et al., 2025) simulate multi-hop logics by linking *offline* static web pages into graphs and constructing questions along random paths (see Figure 8), but their scope remains limited.

Our analysis (Table 1) reveals two critical gaps in current resources. **First**, agent solutions in real-world contexts require accessing and synthesizing information from a dynamic, heterogeneous web—including diverse domains, file processing (e.g., parsing PDFs, CSVs) or active interactions with dynamic web elements (e.g., form submissions, JavaScript-rendered content)—far beyond the static, pre-collected page sets most methods employ. **Second**, existing datasets prioritize pure information seeking, overlooking the need for complex aggregation: 30.79% of WebWalkerQA (Wu et al., 2025b) tasks are solved by simple text parsing, while cases demanding deeper analysis are rare. As shown in Figure 2, robust web agents must not only find relevant facts but also perform deep analysis by aggregating and reasoning with information, *refining gold from sand*. Thus, promoting and evaluating aggregation ability is a crucial but underexplored challenge in web agent research.

To bridge these gaps, we introduce a method (see Figure 2) for automatically constructing training data that requires both information seeking from diversified sources and sophisticated aggregation logics for generalist web agents. This approach adopts **Proactive Online Web Exploring** and **Complex Aggregation Logic Injection**, treating the entire task composition process as an agent-driven pipeline. The agent is equipped with advanced web tools enabling search, static parsing, dynamic interaction, file processing, and vision input, thereby supporting diverse user scenarios, as shown in Appendix A.

Through Proactive Online Web Exploring, the agent collects comprehensive resources beginning from the anchor URL, with domain and complexity regulated by anchor selection and traversal budget. Subsequently, the Complex Aggregation Logic Injection imposes rigorous requirements for both seeking and aggregation. We define a taxonomy of high-level aggregation logics, Element, Set, Scientific Analysis, and Temporal Reasoning, with a total of 12 subtypes (e.g., Math, Statistics) informed by prior studies of multi-hop analysis (Chang et al., 2022; Yang et al., 2018; Talmor et al., 2021; Wu et al., 2025c) and logical reasoning (Ren et al., 2020; Fang et al., 2024). Our analysis shows broad diversity and complexity of aggregation operations in synthesized tasks (Figure 4).

Following quality control, we compile the **WebAggregatorQA** dataset consisting of approximately 10K query-answer pairs, and a challenging human-annotated test set. We further employ rejection sampling on GPT-4.1 sampled trajectories on the constructed dataset, and train our WebAggregator

| Resource | IS | IA | Train set | Information Source | #Domain |
|---|---|---|---|---|---|
| GAIA (Mialon et al., 2023) | ✓ | ✓ | N/A | Text, Figure, Audio, File, Dynamic Web Elements | - |
| BrowseComp (Wei et al., 2025) | ✓ | ✗ | N/A | Text | 9 |
| WebWalker (Wu et al., 2025b) | ✓ | ✗ | Y | Text | 4 |
| TaskCraft (Shi et al., 2025) | ✓ | ✗ | Y | Text, PDF, Figure | 8 |
| WebShaper (Tao et al., 2025) | ✓ | ✗ | N | Text | 11 |
| WebAggregatorQA (Ours) | ✓ | ✓ | Y | Text, Figure, File, Dynamic Web Elements | 12 |

Table 1: Comparison between our WebAggregatorQA created by our method and previous data resources. **IS**: information-seeking, **IA**: information-aggregation. Our method could construct data that covers diverse aggregation needs (Table 6) compared with samples of previous work (Figure 8).

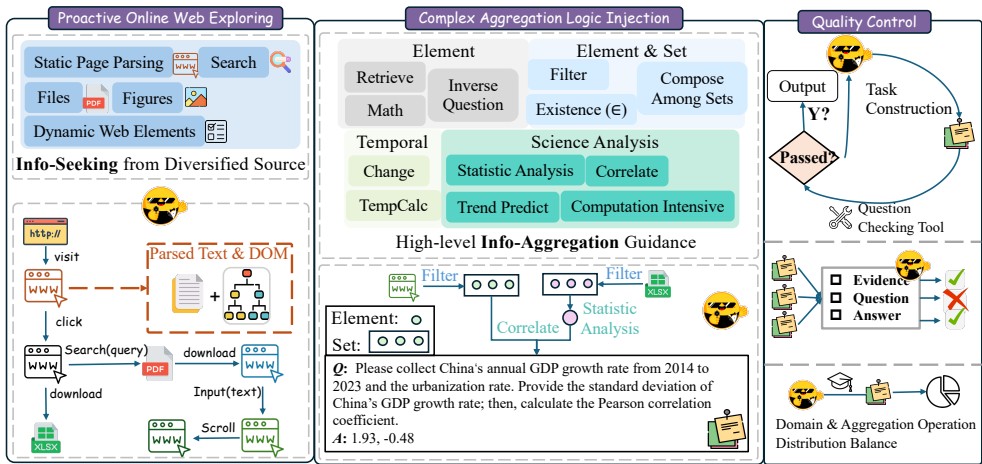

Figure 2: **Data construction pipeline of WebAggregatorQA.** (1) Proactive Online Web Exploring gathers comprehensive information by interacting with the web environment through tools (more details in Figure 9). (2) Task Construction via Complex Aggregation Logic Injection synthesizes QA pairs based on the collected knowledge by instantiating the high-level aggregation guidance into concrete operations, e.g., *Statistic Analysis → standard deviation*. (3) Quality Control ensures the data quality and diversity.

model family based on the SmolAgents (Roucher et al., 2025). Extensive experiments demonstrate that WebAggregator outperforms strong baselines on GAIA-text and WebAggregatorQA, exemplifying the value of our data construction pipeline. The contribution of our work is as follows:

- We propose an automated and scalable workflow for web agent dataset construction, uniquely emphasizing aggregation complexity. The resulting WebAggregatorQA dataset covers a broad range of domains, source types, tool uses, and especially aggregation logics.

- The trained foundation models for web agents, WebAggregator, show superior performance. The WebAggregator-8B surpasses GPT-4.1, and the 32B version surpasses current strong baselines.

- Our test set remains challenging, with even the Claude-3.7-sonnet achieving only 28.3%. Notably, accurate reference retrieval does not guarantee success in aggregation, highlighting the crucial need for progress in this capability.

## 2 WEBAGGREGATORQA

Our objective is to automatically generate at scale a diverse and challenging set of QA pairs grounded in real web resources, suitable for training web agents with *few human involvement*. To reflect realistic scenarios, our tasks require complex information retrieval, deep aggregation, and structured reasoning rather than simple fact lookup.

To achieve this, we propose an automatically verifiable training data construction method illustrated in Figure 2. We frame data synthesis as a web agent task: starting from an anchor URL, an agent performs **Proactive Online Web Exploring** to collect relevant information across heterogeneous sites and file types, then generates QA pairs requiring complex aggregation and reasoning through **Complex Aggregation Logic Injection**. A rigorous automated quality control stage filters out low-quality samples. The process requires only anchor URLs as input, and no human annotation is needed. The agent we used is depicted in Appendix A, including action and observation space.

### 2.1 ANCHOR URL COLLECTION

Diversity of anchor URLs is crucial for broad domain coverage. By selecting seed URLs across multiple domains, we can effectively regulate the dataset's domain distribution, thereby enabling precise control during data construction. We sampled 5,000 topic-diverse queries from QA and scientific datasets (Yang et al., 2018; Jin et al., 2019; Trivedi et al., 2022) and retrieved URLs via Google Search, resulting in over 160,000 anchor links from 11+ domains (see Figure 3).

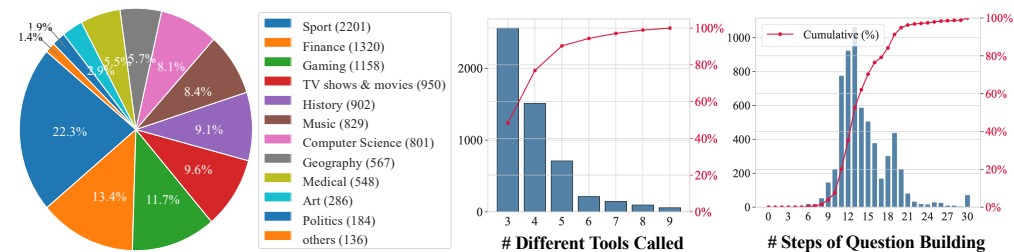

Figure 3: The distribution of domains in WebAggregatorQA, tasks categorized by the number of different tools involved during construction, and steps (an action-observation round) in data synthesis.

## 2.2 DATA SYNTHESIS AS AN AGENT TASK IN REAL INTERNET

In this section, we introduce our approach to synthesizing target QA pairs by framing *task synthesis as a specialized form of web-agent tasks*. The agent is given a task prompt consisting of two components: Proactive Web Exploration and Complex Aggregation Logic Injection, enabling it to complete the task construction in an end-to-end manner.

### 2.2.1 INFORMATION COLLECTION VIA PROACTIVE ONLINE WEB EXPLORING

The first step, **Proactive Online Web Exploring**, aims to collect diverse information as the foundation for task construction. During this phase (Figure 2), the agent is prompted to start from a single anchor URL and employ various tools to navigate across web pages just like human browsing, to discover unknown but relevant information that serves as the basis for generating QA pairs. Interactions include navigating heterogeneous content types such as text, files, and images, as well as dynamic element interactions. To control task difficulty and ensure the comprehensiveness of the knowledge scope, a minimum number of web page visits (e.g., at least $N = 7$) is enforced (see Appendix B.3).

We found that this step could incorporate diversified information from multiple sources. By analyzing the tool calling statistics of 5,296 web exploring trajectories in Figure 3, we found that the proactive web exploration of these tasks involves at least three tools: the Search, Visit, and the compulsory tool ScreenShot. Moreover, 48.36%, 28.55%, and 13.41% of the samples involve the use of 3, 4, and 5 different tools, respectively. The broad interactions here promote greater knowledge diversity and introduce additional challenges—for example, questions derived from file-based information also evaluate the file-processing capabilities of the responding agents.

### 2.2.2 DATA SYNTHESIS VIA COMPLEX AGGREGATION LOGIC INJECTION

The Complex Aggregation Logic Injection procedure aims to incorporate information aggregation behavior requirements into the QA pairs. Hence, the constructed training resource could be expected to enhance the agent's abilities in careful reasoning to deliver concise but valuable insights spawned from the retrieved knowledge, rather than a batch of entities or numbers.

To enhance the diversity of information aggregation, we first define a set of high-level logical operations compiled from human annotations and prior work (Sen et al., 2022; Talmor et al., 2021; Wu et al., 2025c; Fang et al., 2024; Krishna et al., 2025). As shown in Figure 2, aggregation operations are categorized into four major types, Element, Set, Scientific Analysis, and Temporal Reasoning, with a total of 12 subtypes. Element and Set operations are the basis of regular aggregation behaviors among knowledge snippets, such as *mathematical calculation* among elements and *set merging* among sets. While Scientific Analysis and Temporal Reasoning are advanced applications of them, which are expected to reflect user cases and increase the task complexity. These subtypes represent high-level guidance that appears in the prompt to instruct the agent to translate them into concrete reasoning steps, rather than rigid constraints. A vivid example is that math calculations between elements could be derived into addition, subtraction, etc., which could be observed in Figure 4. More detailed seed operations and corresponding prompts are provided in Appendix B.3.

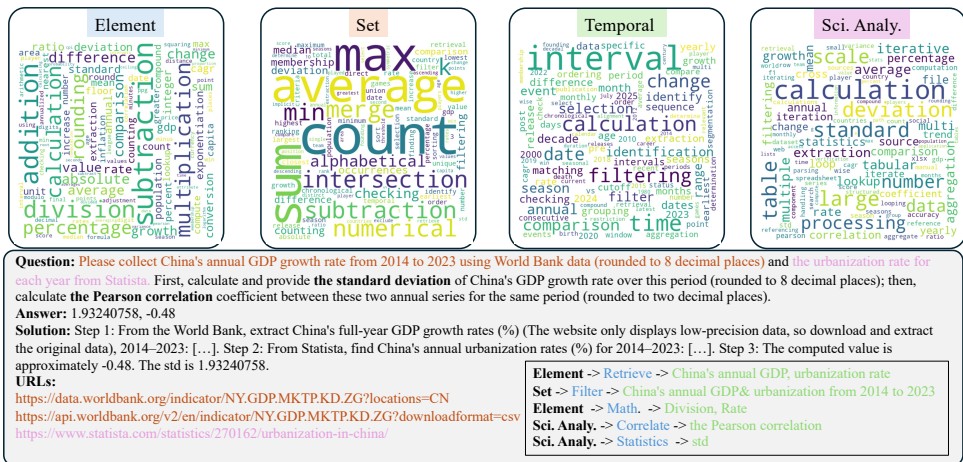

Figure 4: Word cloud of aggregation operations extracted from the constructed tasks. In the **Complex Aggregation Logic Injection** stage, the agent converts high-level guidance into concrete low-level operations to combine knowledge snippets into new conclusions. The illustrated task requires seeking knowledge by `Search`, `Visit`, `Click`, `FileRead`, and aggregations to derive the final answer.

### 2.2.3 QUALITY CONTROL

**QA Alignment Checking**: We implement a two-stage refinement process. First, a self-refinement tool for the agent with a checklist verifies and revises questions before outputting the sample (Appendix B.5). Second, a data checking agent thoroughly reviews the entire task by verifying reference URLs to ensure alignment among questions, answers, and sources (Appendix B.4). About 11.72% of the original data are filtered out in the second stage.

**Diversity Constraint**: We ensure dataset diversity by balancing domain and aggregation operation distributions. First, we annotated anchor URL domains with GPT-4.1 and balanced data to achieve a more balanced distribution (Figure 3). Second, we analyzed information aggregation types using GPT-4.1 to identify low-level operations. Although not perfectly reliable without solving the questions, operations like *calculating average* can be easily detected. We then adjusted prompts to emphasize rare aggregation types, increasing their sample frequency. The word cloud of the aggregation operations (Figure 4) exhibits that different high-level aggregation guidance will spawn diversified low-level, specific operations, e.g., *intersection* for Set, *table* processing for Scientific Analysis.

### 2.2.4 TRAJECTORY SAMPLING

After the task synthesis, we collect the trajectory that completes these tasks. We utilize the agent based on GPT-4.1 with SmolAgents, equipped with almost the same tools exhibited in Table 7, except for the `Screenshot` and `Scroll`, because we only collect the plain text trajectories. To ensure the quality of the collected trajectories, we conduct a further filtering procedure and finally collected 6,184 trajectories for the foundation model training:

- *Correctness* We employ rejection sampling to retain those trajectories that with correct answers according to the reference answers in the WebAggregatorQA.

- *Format* Data with output format errors (e.g., undefined tool name or parameters) is filtered out.

- *Exception Handling* Anomalies in observations (e.g., page failures) are kept to improve the model's generalization, since similar situations would occur in real web environments.

### 2.2.5 STATISTICS OF WEBAGGREGATORQA

WebAggregatorQA comprises 9,883 tasks (with 200 reserved for testing), covering 54,064 unique URLs across 12 domains. Figure 3 shows the distribution of domains and steps for both QA construction trajectories. Domains are labeled by GPT-4.1. Most QA pairs are constructed with around 15 steps, demonstrating that the generated data points are not hastily created from only a few reasoning steps, thus avoiding overly simplistic questions.

## 2.3 Curation of WebAggregatorQA Test Set

Evaluating web agents is vital for their improvement. Existing benchmarks (Wu et al., 2025b; Wei et al., 2025) mainly focus on information-seeking tasks (Figure 8), like deducing answers from ambiguous clues and retrieving entities, often corresponding to *Element-> Retrieve / Inverse Questions* and *Set->Filtering*. 30.29% of WebWalkerQA tasks require only direct retrieval of a single entity, with almost none involving large-scale computation or analysis for the answers.

While this is important for evidence retrieval, the deeper analytical capabilities, such as generating clear and structured answers through reasoning and aggregation (Mialon et al., 2023; Krishna et al., 2025) of evidence, are inadequately evaluated. To bridge this gap, we developed the WebAggregatorQA test set to comprehensively measure both complex retrieval and aggregation skills.

**Annotation Details** We uniformly split 200 tasks as seeds from WebAggregatorQA across different domains to ensure high task diversity. Since humans have inherent cognitive limits in creating highly complex tasks spanning multiple domains (Chen et al., 2025).

**> Step 1:** Human annotators review the QA pairs and references to eliminate ambiguities and provide a revised version of the original data. Our analysis, aligned with prior work (Wei et al., 2025), shows that while questions are generally well-structured, they might lack a unique ground truth due to the high uncertainty of the web. Thus, we ensure every question is unambiguous with exactly one correct answer by adding constraints, e.g., explicit reference sources (the World Bank in Figure 4).

**> Step 2 & 3:** To further enhance sample reliability and reduce bias from the solver's perspective, this process is repeated twice: tasks are solved, ambiguities identified, and revisions made by annotators.

**> Step 4:** In the final cross-validation stage, each question was answered by two annotators, yielding 155 consistently aligned samples. Additionally, there are 4 samples that annotators abandoned during the answering process due to difficulty, but whose references and questions were verified to ensure data quality and thus were retained. More details are shown in Appendix B.2.

This yielded 159 samples, including those in text and multimodal, categorized by difficulty into Level 1 (24), Level 2 (99), and Level 3 (36). Each sample contains a question, reference answer, solution, and supporting URLs. A text example is shown in Figure 4 and a multimodal one is in Figure 10.

## 3 Experiments

### 3.1 Experimental Setups

**Models and Benchmarks** We construct the WebAggregator models by SFT Qwen2.5-7B, Qwen2.5-32B (Yang et al., 2024), Qwen3-8B, and Qwen3-32B (Yang et al., 2025) on the training set of WebAggregatorQA. We evaluate the baselines and our methods on the subset of 103 text-only cases of GAIA (Mialon et al., 2023) following Li et al. (2025a;b); Wu et al. (2025b), and WebAggregatorQA.

**Training Configs** We formalize the trajectory we sampled as (question, $a_1, o_1, ..., a_n, o_n$, answer). $a_i$ stands for the action code the agent generated to perform tool calling, and $o_i$ is the observation returned by the web environment. The question and observations are masked during training.

**Baselines and Metrics** We mainly compare WebAggregator with three types of prior works. **a.** Non-agentic foundation models that answer questions using their internal knowledge. **b.** Zero-shot foundation models initialized as agents via the SmolAgents framework. **c.** Strong fine-tuned foundation models: WebThinker (Li et al., 2025b), WebDancer (Wu et al., 2025a), CognitiveKernel-Pro (Fang et al., 2025), WebSailor (Li et al., 2025a) and WebShaper (Tao et al., 2025). We utilize the pass@1 for performance comparison. The correctness is evaluated by GPT-4.1 with the prompt following previous works (Wu et al., 2025a). Due to the inevitable network fluctuations and CAPTCHA, the agent will be allowed up to two additional attempts when encountering exceptions.

### 3.2 Experiment Results

**Effects of WebAggregatorQA Training Set** The experiment results are shown in Table 2. For the zero-shot foundations, the closed-sourced models surpass the Qwen models on both the GAIA-text

| Methods | GAIA-text | | | | WebAggregatorQA | | | |
|---|---|---|---|---|---|---|---|---|
| | level-1 | level-2 | level-3 | Avg. | level-1 | level-2 | level-3 | Avg. |
| *Non-Agentic* | | | | | | | | |
| GPT-4.1 | 10.3 | 13.5 | 8.3 | 11.7 | 15.4 | 4.0 | 2.8 | 5.6 |
| Claude-3.7-sonnet | 35.9 | 17.3 | 0.0 | 22.3 | 18.5 | 5.1 | 2.8 | 6.8 |
| Qwen2.5-7B | 12.8 | 3.8 | 0.0 | 6.8 | 4.2 | 1.0 | 0.0 | 1.3 |
| Qwen2.5-32B | 20.5 | 9.6 | 8.3 | 13.6 | 4.2 | 1.0 | 0.0 | 1.3 |
| Qwen3-8B | 12.8 | 3.8 | 0.0 | 6.8 | 4.2 | 1.0 | 2.8 | 1.9 |
| Qwen3-32B | 17.9 | 3.8 | 0.0 | 8.7 | 8.3 | 1.0 | 0.0 | 1.9 |
| *Zero-shot Foundations* | | | | | | | | |
| GPT-4.1 | 51.3 | 44.2 | 16.7 | 43.7 | 62.4 | 22.2 | 11.1 | 25.8 |
| GPT-5 | 71.8 | 57.7 | 33.3 | 60.2 | 58.3 | 24.5 | **22.2** | 28.9 |
| Claude-3.7-sonnet | **74.4** | **55.8** | 33.3 | **60.2** | 66.7 | 25.3 | 11.1 | 28.3 |
| Qwen2.5-7B | 23.1 | 15.4 | 0.0 | 16.5 | 27.3 | 3.4 | 2.8 | 6.3 |
| Qwen2.5-32B | 46.1 | 21.2 | 0.0 | 28.2 | 25.0 | 10.1 | 5.6 | 11.3 |
| Qwen3-8B | 33.3 | 11.5 | 0.0 | 18.4 | 30.8 | 5.1 | 5.6 | 9.4 |
| Qwen3-32B | 48.7 | 40.4 | 16.7 | 40.8 | 45.8 | 10.1 | 5.6 | 14.5 |
| *Fine-tuned Foundations* | | | | | | | | |
| *WebThinker* | | | | | | | | |
| Qwen2.5-32B | 56.4 | 50.0 | 16.7 | 48.5 | | — | | |
| *WebDancer* | | | | | | | | |
| Qwen2.5-7B | 41.0 | 30.7 | 0.0 | 31.0 | | — | | |
| Qwen2.5-32B | 46.1 | 44.2 | 8.3 | 40.7 | | — | | |
| *WebSailor* | | | | | | | | |
| Qwen2.5-7B | - | - | - | 37.9 | | — | | |
| Qwen2.5-32B | - | - | - | 53.2 | | — | | |
| *WebShaper* | | | | | | | | |
| Qwen2.5-32B | 61.5 | 53.8 | 16.7 | 52.2 | | — | | |
| *CogKernal-Pro* | | | | | | | | |
| Qwen3-8B | 56.4 | 42.3 | 8.3 | 43.7 | | — | | |
| *WebResearcher* | | | | | | | | |
| Qwen2.5-7B | 53.8 | 30.8 | 16.7 | 40.8 | 37.5 | 11.1 | 8.3 | 14.5 |
| *- pass@3* | 74.4 | 63.5 | 25.0 | 63.1 | 54.2 | 22.2 | 19.4 | 26.4 |
| Qwen2.5-32B | 66.7 | 44.2 | **33.3** | 51.5 | 54.2 | 15.2 | **11.1** | 20.1 |
| *- pass@3* | 79.5 | 67.3 | 50.0 | 69.9 | 70.8 | 22.2 | 19.4 | 28.9 |
| Qwen3-8B | 61.5 | 34.6 | 16.7 | 42.7 | 54.2 | 11.1 | 5.6 | 16.4 |
| *- pass@3* | 82.1 | 53.8 | 33.3 | 62.1 | 62.4 | 21.2 | 11.1 | 25.2 |
| Qwen3-32B | **69.2** | **55.8** | 16.7 | **56.3** | **62.4** | **24.2** | 8.3 | **26.4** |
| *- pass@3* | 79.5 | 67.3 | 50.0 | 69.9 | 66.7 | 35.4 | 13.9 | 35.2 |

Table 2: The Pass@1 & Pass@3 performance of agents on GAIA-text and WebAggregatorQA. The best performance of different settings is in bold.

and WebAggregatorQA. However, after tuning on WebAggregatorQA, Qwen models exhibit clear and steady improvements on GAIA-text and WebAggregatorQA and approach the performance of these strong baselines. Specifically, the WebAggregator based on Qwen2.5-32B and Qwen3-32B surpasses most of the strong baselines, including GPT-4.1 and WebShaper. The pass@3 performance of WebAggregator-32B achieves 69.9 on GAIA-text. These observations prove the quality of WebAggregatorQA and the effectiveness of our data construction paradigm.

**Difficulty of WebAggregatorQA Test Set** WebAggregatorQA poses a new challenge for current agent systems. GPT-4.1-powered SmolAgents attain 43.7% accuracy on GAIA-text but drop to 25.8% on WebAggregatorQA. Claude-3.7-sonnet shows a similar decline. Furthermore, the performance gap between Claude and GPT-4.1 is smaller on WebAggregatorQA than on GAIA-text. This suggests that for the harder questions in WebAggregatorQA, neither model can solve them effectively, which leads to the reduced

| Model | WWQA | XBench |
|---|---|---|
| WebDancer-7B | 36.0 | - |
| WebSailor-7B | - | 34.3 |
| WebAggregator-7B | **44.7** | 37.0 |
| WebAggregator-8B | 41.2 | **40.0** |

Table 3: Performance on XBench and WWQA (WebWalkerQA).

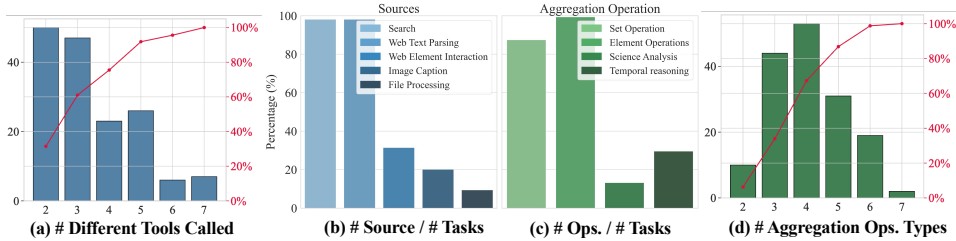

Figure 5: Distributions of tasks required different numbers of tools (a) and aggregation operations (d). Proportion of information source (b) and aggregation operations (c) that are needed across tasks.

gap. These results highlight the substantial gap between current agent capabilities and the demands of information aggregation needed for multi-hop web tasks.

**Transferability of WebAggregator Models**  Considering the response latency and efficiency of small foundation models, it is crucial to further enhance these smaller foundations to offer society a more affordable yet powerful alternative. To explore their potential, we evaluate these models on two additional benchmarks: WebWalkerQA (Wu et al., 2025b) and XBench (Chen et al., 2025), as summarized in Table 3. In Table 2, WebAggregator-8B achieves performance comparable to GPT-4.1 on GAIA-text, demonstrating strong capabilities despite its relatively smaller size. Furthermore, both WebAggregator-8B and 7B significantly outperform previous strong baselines on WebWalkerQA and XBench. Although these results confirm that smaller WebAggregators excel on these benchmarks, they still face challenges with the more difficult tasks in WebAggregatorQA, where WebAggregator-8B notably trails behind the 32B counterpart and GPTA-4.1. Consequently, breaking through the performance bottleneck of small foundation models on hard tasks remains a vital direction.

## 4  ANALYSIS

### 4.1  HOW TO SOLVE WEBAGGREGATORQA

We present the distribution of the information source and aggregation operations needed to solve WebAggregatorQA, as shown in Figure 5.

**Diversified Information Source Reliance**  We observe that all of the tasks of WebAggregatorQA need information from only Search and Web Text. Moreover, the tasks also require information from Files and do not rely solely on one source. Solving WebAggregatorQA requires advanced web-browsing capabilities to retrieve knowledge. The task is highly challenging for models that rely only on their internal knowledge: even strong base models such as Claude-3.7 and GPT-4.1 correctly solve fewer than 7% of the questions. The advantage of GPT-4.1 and Claude over the Qwen series is largely attributable to their multimodal (image) processing capabilities. Without access to tools to fulfill multimodal understanding, Qwen models can only answer a small fraction of questions.

**Diversified Information Aggregation Requirements**  We observe that the information aggregation requirements of WebAggregatorQA challenge the agent systems. Figure 5 illustrates that all of the tasks possess these operations, and many of them contain multiple operations, further increasing the task difficulty. We then further examine the impact of information aggregation. We analyze the agents' trajectories to identify the frequency of a specific failure mode: *successfully retrieving all of the reference URLs but fails the task*. From Table 4, a clear trend emerges: for tasks where agents successfully visit all reference URLs, accuracy is notably higher than the overall average, suggesting that successful information retrieval is helpful for correct aggregation. While Qwen3-32B is a notable exception, indicating its performance bottlenecks may lie beyond simple retrieval. In this context, our WebAggregator-32B not only demonstrates this accuracy improvement but also achieves performance comparable to other strong baselines. Nevertheless, even with complete information, no agent reaches a perfect score, underscoring that the complex aggregation logic remains the ultimate challenge.

| Model | Counts | Acc. |
|---|---|---|
| GPT-4.1 | 30 | 33.3 |
| Claude | 38 | 42.1 |
| Qwen3-32B | 22 | 9.7 |
| WebR-32B | 28 | 35.7 |

Table 4: Counts and accuracy of trajectories that visited all of the reference URLs.

**Retrieval vs. Aggregation Tasks**   We evaluate the impact of information aggregation on agents by analyzing their tool usage patterns across WebAggregatorQA and WebWalkerQA, as shown in Figure 6.

We define tool call density as the percentage of steps that involve tool usage. We observe that while tasks in WebAggregatorQA require more total steps to complete, the tool call density is notably lower. This pattern suggests that in WebAggregatorQA, models rely more heavily on reasoning steps to execute information aggregation—enabling deeper synthesis and analysis of existing information—rather than predominantly invoking tools to acquire new external knowledge.

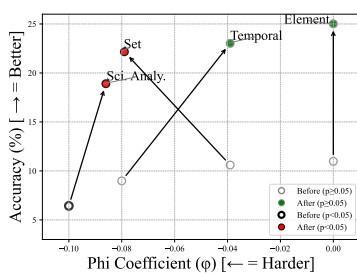

Figure 6: Steps and tool use density of two models across test sets.

### 4.2  BEHAVIOR EVOLUTION OF WEBAGGREGATOR

**Evolution on Tool Usage Behaviors**   To understand how fine-tuning on WebAggregatorQA alters agent behavior, we analyzed its operational steps, classifying them as either tool-calling (information-seeking) or non-tool-calling (reasoning). Our findings, illustrated in Table 5 and Figure 12, reveal three key trends. First, WebAggregator agents exhibit less tool dependency and more reasoning, significantly reducing their tool usage across all model sizes. Second, we found an inverse correlation with scale, where larger models rely on fewer tools. Finally, the agent's behavior demonstrates increased complexity; post-tuning, tool-calling patterns shift from a simple, concentrated structure (e.g., 40% of initial trajectories used only two tools, as shown in Figure 12) to a far more diverse and sophisticated distribution.

**Evolution on Aggregation Capabilities**   We investigate how different aggregation logic types affect model performance, focusing on failure patterns common across models rather than model-specific issues. Logic labels are generated by an LLM, consistent with the dataset construction method (Figure 2). We compare Qwen series models before and after fine-tuning. We compute the Phi Coefficient ($\psi$), measuring correlation between logic type presence and answer correctness. More negative $\psi$ indicates greater difficulty. Figure 7 reveals that computation-intensive logic, particularly Scientific Analysis and Set Operations, remains the key performance bottleneck even for WebAggregators.

| Model | %tc steps | $\mathrm{sum(tc)/step}$ |
|---|---|---|
| Qwen3-8B | 90.27% | 1.52 |
| +WAgg | **86.37%** | **0.94** |
| Qwen3-32B | 91.55% | 1.12 |
| +WAgg | **82.14%** | **0.89** |

Table 5: The tool calling behaviors of Qwen3-8B and Qwen3-32B on WebAggregatorQA before and after training.

**Training Efficiency**   The construction of datasets and the training of web agent models are typically resource-intensive processes. If satisfactory performance can be achieved with a smaller trajectory size, the approach would become more accessible and cost-effective. Motivated by this, we train the Qwen3-8B on small subsets of WebAggregatorQA, using 500 and 1,200 samples, respectively. The model trained on 500 samples attained 36.9% accuracy on GAIA-text, while the one trained on 1,200 samples achieved 38.83%. These results further demonstrate the high quality of WebAggregatorQA; even a small-scale subset can deliver significant performance gains for foundation models.

Figure 7: Phi Coefficients ($\psi$) by Logic Type across Qwen series models, before and after Tuning on WebAggregatorQA.

### 4.3  SCREENING ON FAILURE MODES

To understand the challenges posed by WebAggregatorQA, we analyze failure trajectories where the agent successfully retrieved all necessary information but still produced an incorrect answer. Our analysis is based on a manual examination of 50 such cases from models in Table 4. We identify a taxonomy of interconnected failure modes, categorized below.

**Information Seeking** Errors at this stage prevent the agent from accessing or correctly interpreting necessary evidence. This can manifest as a **Source Finding Failure**, where the agent fails to locate the correct webpage due to high uncertainty or being misled by preliminary steps, often settling for secondary sources with incomplete information.

**Information Aggregation and Reasoning** This category covers logical errors during the aggregation of retrieved information. One primary issue is **Faulty Aggregation**, where the agent applies flawed reasoning, such as unsupported assumptions, fails to leverage key information and thus misinterpreting evidence (e.g., Figure D.1). Another failure is **Error Accumulation**, where minor inaccuracies, like premature rounding, compound across steps to yield a significantly incorrect result (Figure D.3). Compounding these issues is **Underperformed Iterative Refinement**, a critical failure where the agent cannot self-correct erroneous steps, causing it to proceed down an incorrect logical path without identifying or revising its mistake(e.g., Figure D.2).

**Agent Foundation Capability Failures** Agents are also prone to operational failures that derail the task. For instance, they may make **Invalid Tool Calls** by providing improper arguments (e.g., a poorly formulated query or an invalid URL), which often yield empty results and waste valuable steps. Such inefficiencies can lead to **Exceeding Max Steps**, for the agent becomes trapped in error-correction loops or simply wastes of steps due to poor planning, ultimately failing to complete the task.

## 5 RELATED WORK

**Resources for Web Agent Foundation Models** Multi-hop questions for training web agent foundation models (Tongyi, 2025; Qiao et al., 2025) require advanced tool use, complex reasoning, and grounding in real-world web environments, making manual dataset construction challenging. Existing QA datasets, such as HotpotQA (Yang et al., 2018) and Musique (Trivedi et al., 2022), do not capture the intricacy of authentic web interactions. While some works generate request-action pairs (Xu et al., 2025; Chen et al., 2024; He et al., 2024), these are not applicable for goal-oriented web tasks. Recent methods first construct logical flows over knowledge snippets and then synthesize tasks accordingly (Wu et al., 2025a; Li et al., 2025a; Shi et al., 2025; Tao et al., 2025; Xia et al., 2025). For instance, knowledge graphs built from *offline* pages are used for task generation (Shi et al., 2025; Wu et al., 2025a), and entity expansion or formalization helps model logic flows (Xia et al., 2025; Tao et al., 2025). However, these approaches are restricted by their dependence on static pages and often neglect the aggregation of information from diverse sources (Figure 2). Moreover, their complexity mainly comes from entity tracing rather than synthesizing information across multiple sources.

**Benchmarking Web Agents** Most existing benchmarks focus on information-seeking, requiring agents to use tools and perform multi-hop reasoning in realistic web scenarios, as in WebWalker (Wu et al., 2025b) and BrowseComp (Wei et al., 2025). Few research (Li et al., 2025c) assess information aggregation. FRAMES (Krishna et al., 2025) aim to evaluate the factuality, retrieval, and aggregation abilities of LLMs, but their knowledge scope is limited to Wikipedia. WideSearch (Wong et al., 2025) addresses aggregation by constructing tasks involving many simple actions. GAIA (Mialon et al., 2023), which is most relevant to our work, evaluates general capabilities with human-constructed tasks. However, recent agents (Fang et al., 2025; Qiu et al., 2025) perform well on GAIA, indicating a crucial need for more challenging benchmarks that jointly evaluate information-seeking and aggregation.

## 6 CONCLUSION

In this work, we identify the critical limitation of existing web research agents, their inadequate focus on information aggregation, which restricts their capacity for generating insightful and coherent research outputs. To address this, we propose an automated, agent-driven data construction paradigm that enables the synthesis of diverse and verifiable tasks demanding both information seeking and complex aggregation across real-world web environments. Our resulting WebAggregatorQA dataset and the foundation model family, WebAggregator, demonstrate substantial improvements over current baselines on GAIA-text and WebAggregatorQA. Notably, even advanced commercial models like GPT-4.1 and Claude-3.7-sonnet struggle on these tasks. Even after retrieving all of the references, the agents still struggle on WebAggregatorQA, reflecting the importance and difficulty of effective information aggregation for web agents.

ETHICS STATEMENT

In this paper, we do not involve any significant ethical issues. First, the data we construct is based on URLs obtained through information retrieval from existing datasets, which do not contain sensitive content such as violence. All datasets and models used in this work strictly follow the licenses specified by their original authors. Additionally, this paper involves manual annotation; detailed explanations of the annotation process and the remuneration provided to annotators are given in Appendix B.2.

REPRODUCIBILITY STATEMENT

We will provide comprehensive details to facilitate the reproducibility of our work. We commit to open-sourcing all relevant code, including the agent workflow, data construction, and evaluation scripts. The constructed dataset—comprising pure QA pairs, trajectories, and URL collections—will also be made publicly available. Furthermore, the trained model checkpoints, the qwen3-8B and 32B versions of WebAggregator, will be released alongside inference code, which includes benchmark evaluation and a quick-start guide.

Currently, we have already uploaded the agent workflow, data construction, and evaluation scripts; samples of training set of WebAggregatorQA due to space limitation and the whole testset in the **Supplyment Materials**.

USE OF LLMS

In this paper, we mainly use LLMs as grammar checking tools after our authors finished the manuscript. LLMs are not involved in primary manuscript writing, coding, and other tasks.

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

| Operations | Questions |
|---|---|
| *Element Operations* | *Aggregate elements/entities, e.g., numbers, times, names$(x, y)$.* |
| Retrieve $(x)$ | In Amor: A Recipe for Building Adaptable ... , what hourly pay (in USD) is for the hired NLP expert? |
| Inverse $(x)$ | Which American actor won the Academy Award for XXX in the 1990s released their first solo studio album the greatest number of years after their Oscar win? |
| Math $(x, y)$ | Among Benedict, Robert Downey, and ..., for the person's first appearance in a Marvel Cinematic Universe film corresponded to the highest ROI for their debut Marvel movie, what is the ROI (three decimals)? |
| *Set Operations* | *Aggregate elements $(x)$ and sets $(Y, Z, ...)$.* |
| Filter$(Y)$ | Among the countries that won at least 15 gold medals at the London 2012 Summer Olympics, what is the HDI of the country that had the third highest per-capita GDP (in USD) in 2012? |
| Existence$(x, Y)$ | For the college that had the most players selected overall in the 2023 NBA Draft, how many of its draftees were picked in the first round? |
| Compose$(Y, Z)$ | According to the WorldPopulationReview, how many cities among the top 100 most populous cities in 2025 have experienced a population decrease compared to 2024? |
| *Temporal Reasoning* | *Reasoning or calculation related with time.* |
| Change | Between 1990 and 2022, which country had the third largest average annual percentage increase in nominal GDP? |
| TempCalc | Among Robert De Niro, Al Pacino, Christopher Walken, and Jessica Lange, who has the longest interval between their first and most recent Academy Award nominations without winning, and what is the length of that span in years? |
| *Science Analysis* | *Coding is a must to improve efficiency or precision for computation-intensive tasks.* |
| CompIntensive(X) | What is the average closure price of Apple.inc from Jan. 2024 to Oct. 2024? |
| Predict $(x_1, ..., x_n)$ | KFF published an article on abortion in Women's Health Policy on Feb 27, 2025. Using single exponential smoothing and MSE, search for the optimal alpha (0.01-0.99, step=0.01) based on the historical data, the MSE loss, and use the alpha to estimate the next data point. |
| Statistic $(x_1, ..., x_n)$ | Among all Cleveland Cavaliers head coaches who have won at least one playoff game with the team, what is the standard deviation of their playoff win percentages? |
| Correlate $(X, Y)$ | Between the 2012 to 2022 NBA seasons, what is the Pearson correlation coefficient between Damian Lillard's season average points per game and the Portland Trail Blazers' regular season win percentage? |

Table 6: Several representative examples in WebAggregatorQA of information aggregation operations are presented. Note that the operations here are *high-level guidance* that could be derived into a diversified, specific form, rather than low-level constraints. $x$ means an element or knowledge snippet, $X$ denotes a list of knowledge snippets that fulfill a certain condition.

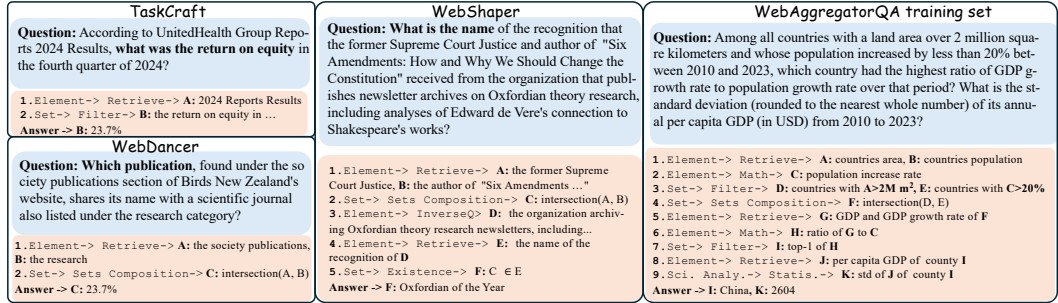

Figure 8: Samples from TaskCraft (Shi et al., 2025), WebDancer (Wu et al., 2025a), and WebShaper (Tao et al., 2025) primarily evaluate basic information-seeking skills, such as *Element -> Retrieve* and *Set -> Sets Composition* for entity filtering. In contrast, the selected WebAggregatorQA samples demand significantly more complex information aggregation to derive final answers. Crucially, these diverse aggregation strategies are ***automatically constructed*** by agents, guided by high-level logics and accumulated knowledge during data construction, resulting in rich variability that reflects task-specific intricacies.

Figure 9: A running example of **Proactive Web Exploring**: a greater variety of interactions fosters a richer diversity of knowledge and introduces more challenges throughout the process, e.g., questions built from file knowledge also test the file-processing abilities of responding agents.

## A  AGENT STRUCTURE

First, we introduce our agent framework. User instructions often require accessing diverse information on the web—plain text, images, or files—some needing specific interactions like button clicks. Thus, the agent must go beyond extracting static page text. We categorize tools by information source: **Search** (Search), **Static Page Parsing** (Visit, StrFind), **Dynamic Interaction** (Input, Click, Scroll, Goback), **File Processing** (FileRead), and **Image Captioning** (ImageCaption).

Our implementation utilizes a code-based ReAct (Yao et al., 2023) agent, built on SmolAgents (Roucher et al., 2025), which outputs natural language thoughts, Python-coded actions,

| Action | Observation |
|---|---|
| Search(Query) | Search results |
| Visit(URL) | Web Text & DOM |
| StrFind(Query) | Matched str in text |
| Input(str, tbox_id) | Web Text & DOM |
| Click(button_id) | Web Text & DOM |
| Scroll(Pixels) | Web Text & DOM |
| Goback() | Web Text & DOM |
| FileRead(Path) | File content |
| Screenshot(Path) | Capture screen |
| ImageCaption(Path) | Image description |

Table 7: Action and Observation space.

and receives code log as environment observations. Each task runs within a fixed 30-step budget, where a step includes an agent action and its observation. We extend the *opendeepresearch* SmolAgents instance with DOM parsing for finer web interactions (see Table 7). This web agent effectively handles tasks requiring information from multiple sources, as summarized in Table 1.

## B  MORE DETAILS FOR WEBAGGREGATORQA

### B.1  MORE EXPLANATIONS OF QUALITY CONTROL

**QA Alignment-based Filtering**  To improve data quality, we implemented a two-stage refinement process for task construction. The first stage uses a self-refinement tool with a checklist (Appendix B.5) to quickly verify and revise questions during creation. Items meeting all criteria are accepted and outputted; those that don't are revised based on feedback until they comply. The second stage involves a data checking agent that thoroughly reviews all reference URLs to ensure alignment of question, answer, and sources (Appendix B.4). About 11.72% of the data were filtered out for failing to meet these standards.

**Diversity Constraint**  We ensure dataset diversity by balancing domain and aggregation operation distributions. First, we annotated anchor URL domains with GPT-4.1 and sampled data to achieve a more balanced domain distribution, shown in Figure 3. Second, we analyzed information aggregation types using GPT-4.1 to identify high- and low-level operations (e.g., *math subtraction*). Although

not perfectly reliable without solving the questions, common operations like *calculating average GDP* can be accurately detected. We then adjusted prompts to emphasize rare aggregation types, increasing their sample frequency. Figure 4 shows a word cloud illustrating how different high-level aggregation tasks yield diverse specific operations, such as *intersection* for set operations and *table* processing for Science Analysis.

**Data Leakage Avoidance**   During proactive web exploration, agents may download and parse existing datasets. To prevent data contamination, we created a website keyword blacklist. Pages matching the blacklist or containing identified datasets were excluded from retrieval and subsequent model evaluation to ensure the fairness of the evaluation.

## B.2    TESTSET ANNOTATION

### B.2.1    DATA COLLECTION AND VERIFICATION

The construction of the test set includes the seed tasks collection, several rounds of revision, and cross-validation procedures. Three human annotators are involved, all of them with at least a bachelor's degree. Each sample requires an average of 3 hours of annotation work, and the whole procedure lasts for more than 4 weeks of part-time work (4 hours a day).

**Seed Tasks Collection**   A single human annotator, even if highly specialized in one domain, faces inherent limitations in generating diverse and comprehensive samples across multiple fields. To address this, we engage multiple annotators to revise 200 topic-diverse tasks, uniformly selected from WebAggregatorQA based on their domain labels. These carefully revised seed examples help ensure that the test set attains the desired diversity.

**Task Revision Principles**   Our initial analysis, consistent with prior work (Wei et al., 2025), shows that due to high uncertainty in the web environment and an answer-to-question task design, questions are generally well-structured but often lack unique ground truths. While this ambiguity may be tolerable for training, it is unacceptable for testing. Thus, our key revision principle is to ensure each question is unambiguous and has exactly one correct answer.

• **QA** (1) Ensure clarity in the question statements. All claims must be explicitly stated, and if multiple sources of evidence exist, additional constraints should be provided in the question to avoid ambiguity. (2) The reference answer must be the sole feasible and correct one to the question.
• **Reference** Reference information, including URLs and solutions, is vital to the revision process. When these reference URLs and solutions are properly validated, the quality and reliability of the questions and answers are assured. Accordingly, annotators are required to: (1) verify the reliability of URLs, ensuring they originate from authoritative and reputable sources; (2) ensure consistency: the evidence remains stable and not prone to variation across different websites, contexts, or over time; (3) confirm the fidelity of URLs: each provided reference URL directly and substantively supports the question. Those pages that have a strict CAPTCHA will be replaced with more stable ones. Then the questions and answers are revised accordingly.

The second principle is to **increase task complexity** from the same two perspectives: complex information aggregation and diversified information sources. We provide annotators with the information aggregation guidance and encourage them to incorporate more reasoning steps into the questions to enhance their difficulty. They are also advised to leverage various information forms beyond plain webpage text. The answer should not be directly found on the web page.

**Verification**   We utilize agents to assist the human validation procedure. Initially, a GPT-4.1-powered agent attempts to solve the questions, facilitating identification of potential ambiguities from the solver's perspective within a realistic web environment. Subsequently, human annotators review the agent's responses, detect any misunderstandings, and revise the questions, solutions, and answers accordingly. This cycle is repeated once more to ensure comprehensive disambiguation.

In the final stage, human annotators independently solve the tasks for cross-validation purposes. Samples that fail to achieve consensus with previously annotated answers are excluded. The independent annotation achieved agreement with 155 out of 159 references, with 4 tasks omitted due to excessive complexity, thereby validating the quality of the references.

> **Question:** Between the game's release month and three months afterward, what was the average monthly percentage change in peak concurrent players for a superhero PvP shooter game released in 2024 (from steamcharts.com)? The game lost the highest average number of players in a month before July 2025. At the beginning of that month, there was a Twitch Drops event where watching for 30 minutes rewarded an item featuring two characters. Before July 2025, how many times was the character on the right buffed and nerfed, respectively? Rounded to two decimals.
>
> **Answer:** -2.78; buffed: 3, nerfed: 7
>
> **Solution:** 1. Get peak concurrent players for Marvel Rivals for Dec 2024, Jan 2025, Feb 2025, and Mar 2025 from statistics (…).
> 3. Average the percentages: -2.78%. 4. The month this game lost the highest number of average players is March 2025.
> 5. The first twitch drops shown in marvelrivals.com is 20250217.
> The 30mins reward is a spray, the human torch on the right of the spray.
> 6. Finally, we can count the ↓ (nerf), 7, and ↑ (buffed), 3.
>
> **URLs:**
> https://steamcharts.com/app/2767030
> https://www.marvelrivals.com/announcements/20250217/40955_1212338.html
> https://marvelrivals.fandom.com/wiki/Human_Torch#Balance_Changes

Figure 10: A multimodal sample from the test set of WebAggregatorQA. To solve this task, the agent must extract information from the image to obtain clues for the next step. Since the image is not provided with the question, the agent is required to locate the relevant picture independently.

### B.3 DATA CONSTRUCTION PROMPT

> **Part-1**: Proactive Online Web Exploring
>
> ```
> URL:
> {URL}
>
>
> --
> Task Overview
> • Create a challenging multi-hop question based on the given URL
> and related information.
>
> • Ensure the quality of the answer when providing a reference
> answer! Please calculate and verify the reference answer before
> giving the final data.
> • The question should be written in the SAME language as the
> website content.
> --
> 1.  Information Gathering
> • Start by thoroughly exploring the given URL and its description.
> • Visit and browse at least **{least_visits} different websites**
> to collect diverse and relevant information.
> • Avoid relying solely on simple search engine queries or Wikipedia.
> Instead, actively browse, jump between pages, and record your
> navigation steps and key findings.
> • After each browsing action, briefly document what you did and the
> important information you discovered.
> --
> 2.  Question Design
> Formulate a **multi-hop question** that MUST requires reasoning
> across multiple sources.  The answer should **not** be obtainable
> by a simple search or from a single page.
>
> The question should be:
> • Challenging but natural and concise, as if a real user is seeking
> to learn or solve a puzzle.  Avoid unnatural or arbitrary questions
> such as summing unrelated numbers.
>   - e.g., year * (number of countries of china) is unacceptable!
> • Self-contained.
>   - Illustrated with essential clues that guide the respondent
> to locate the information without explicitly naming the sources
> or searching queries.  The clues must be necessary but precise,
> avoiding overly broad candidates.
>     - BAD EXAMPLES: Some China city has,...  (NOT self-contained!
> Specify the city by specifying the name or providing clues.)
> ```

- Based on specific details from at least 5 to 8 different web pages.
- Reflective of the domain's characteristics (e.g., medical: functions, gaming:  guidance, players, chemistry, math, puzzles).

*Part-2*: Complex Aggregation Logic Injection

**3. Composition Reasoning Operations (Mandatory)**
Incorporate at least one of the following reasoning operations in your question:
● **Science Analysis**
> **Statistical Analysis**
  – Analyze data from web pages, you may use, but not limited to: calculating the mean, variance, or standard deviation within a specified time period. Some good examples:
    1. What is the median winnings for drivers who have driven a Chevrolet car?
    2. Which category exhibits the most consistent growth rate across the 5-year period, and what is the average annual percentage increase for that category?
    3. Can you calculate the standard deviation of the average comprehension scores across A, B, and C?
> **Correlation Analysis**
    1. What is the Pearson correlation coefficient (to two decimal places) between China's average annual temperature and its $CO_2$ emissions per capita over the same period?
> **Trend Forecasting**
  – Based on historical data, predict future data points. Any algorithm can be used, such as linear regression, polynomial regression, logistic regression, EMA, etc. REMEMBER: Clearly specify the basis for prediction to ensure a unique answer. Some good examples:
    1. Considering the historical data from 1961 to 1967, what could be the forecasted points of Suzuki in the 50cc and 125cc classes for the upcoming years? Use the average growth rate or the most recent 5-year growth rate for prediction.
    2. KFF published an article on abortion in Women's Health Policy on Feb 27, 2025. Using single exponential smoothing and MSE, search for the optimal alpha (0.01-0.99) based on the historical data, the MSE loss, and use the alpha to estimate the next data point.
> **General Computation Intensive Tasks**
  – Batch Data Analysis Requires Intensive Computation. The need to retrieve and process large lists of numbers makes coding ESSENTIAL.
    1. What is the average closure price of Apple.inc from Sep. 2024 to Oct. 2024?
    2. Across all NBA seasons where Manu Ginobili's Player Efficiency Rating (PER) exceeded 20 in the regular season, what was the average number of regular season wins by his team?
> **Other Tasks**
  – Complex Algorithm with high Complexity: Try to design problems that require coding to reduce time complexity.
  ----------------
● **Element-wise operations**
> Calculation
  – Selecting specific elements, performing mathematical operations between elements, e.g., probability, calculation.
  – Examples:
    1. What is the sum of A's speed and B's speed?
    2. By how much does C's value exceed D's value?
    3. What is the difference between the population of city X and city Y?
> Inverse Question
  – Formalized as an inverse question about certain information. Avoid direct listing; use indirect clues framed as questions. Ensure your phrasing uniquely identifies the subject without ambiguity.
  – Examples:

```
    1.  Instead of "Tom is a singer from New York, who was born on
11 Nov 2024, he...", you can use "for the single from New York, who
was born on 11 Nov 2024, he...".
    2.  In June 2022, researchers from Huddersfield University
published a paper on the application of YOLO in agriculture.  My
research primarily focuses on ...
----------------
● List/Set-wise operations
  > sorting (alphabetical, numerical, top-K), sum, average,
counting, intersection, subtraction, merging.  Examples:
    1.  Which is the shortest among XXX?
    2.  What is the average length of YYY?
    3.  How many items appear in both set A and set B?
    4.  What is the total number of Z across all categories?
----------------
● Element-Set operations
  > checking membership or counting occurrences.  Examples:
    1.  Is element E part of the top 10 ranked items?
    2.  Exclude all names that were born in 1984 from ...
    3.  Between 2012 and 2021, was the rate of increase in China's
average annual temperature higher or lower than the global average?
    4.  On the same day that a landmark house on South Main Street
in Coeymans Landing, New York, rich with local history, built
in the late 1830s, officially entered the National Register of
Historic Places listing, how many places entered the list total?
----------------
Note:  The numbers or elements used in these operations should be
discoverable by reading the web content, not directly provided in
the question.
----------------
4.  Answer Requirements  > The answer MUST not be obtained directly
from the retrieved text and MUST be derived through reasoning.  >
Short, Concise and easy to verify.
> Stable over time (avoid dynamic or real-time data).
> Of a clear entity type (e.g., person, number, date, place).
----------------
5.  Output Format
Output your final result in the following JSON format:
{
  "topic":  "Brief description of the question's domain or topic",
  "question":  "The constructed multi-hop question",
  "answer":  "The answer X",
  "context":  {
  "URLs":  [ "url_1", "url_2", "url_3", "url_4", "url_5", ...  ]
  }
}
```

## B.4 PROMPT OF DATA QUALITY CHECKING AGENT

---

**TASK DESCRIPTION OF DATA QUALITY CHECKING AGENT**

{Composition Reasoning Operations Prompt}
Evidence Checking

- ☐ **URL Validity**: Verification that all URLs conform to proper syntax and resolve correctly without errors.
- ☐ **Information Relevance**: Assessment of whether each URL contains information that is necessary and sufficient to address the research question.

Question Checking

- ☐ **Self-Containment**: The extent to which the question is fully specified and comprehensible without requiring additional external context.
- ☐ **Retrieval Necessity**: The degree to which answering the question necessitates consulting external sources, while avoiding excessive disclosure of information within the question itself.
- ☐ **Aggregation Necessity**: The question must include at least three different aggregation operations, ensuring that the answer cannot be obtained through direct retrieval.
- ☐ **Clarity**: The precision and unambiguity of the cues or references embedded in the question that facilitate accurate data retrieval. The clues will not lead to multiple feasible answers.
- ☐ **Temporal Stability**: The property that the correct answer to the question remains consistent over time, unaffected by temporal changes (e.g., "Who was the immediate past president of the United States?").

Answer Quality Assessment

- ☐ **Information Fidelity**: The extent to which all information presented in the reference answer is fully consistent with the URLs or other provided external information sources.
  - *Example of inconsistency*: The temperature retrieved from the reference URL is 37°C, whereas the solution states 35°C, resulting in an erroneous calculation of the average temperature.
- ☐ **Ground Truth Validity**: The reference answer must accurately and unambiguously reflect the requirements of the question, conforming to information obtained from authoritative and reliable data sources.
  - The answer should be derived from recognized authoritative channels or verified databases.
  - Ensuring verifiability through reliable sources is especially important for questions involving numerical data, statistics, or other factual information.
  - *Example of invalid answer*: "The moon's distance from Earth is 100,000 km." This contradicts scientific consensus, which states the distance is approximately 384,400 km.
- ☐ **Uniqueness and Unambiguity**: The reference answer should be uniquely correct, avoiding ambiguity or multiple plausible solutions.
  - Are there conflicting data from multiple sources that lead to multiple possible answers?

---

```
          • Are there precision conflicts between different data
            sources (e.g., 33.2 vs.  33.20987)?
  ---------------
  Based on the above criteria, analyze the following data:
  Question:  {}
  Answer:  {}
  Evidence_URLs:  {}
  Please verify whether each item meets the standards.
  ---------------
  Output Format
  Return your analysis in the following JSON format:
  ```json
  {
    "Evidence Passed":  1 or 0,
    "Question Passed":  1 or 0,
    "Answer Passed":  1 or 0,
    "Domain":  "[USE ONLY ONE WORD OF THE FOLLOWING!] Gaming, Sport,
  TV shows & movies, Computer Science, Art, History, Music, Geography,
  Politics, Finance, Medical, Law",
    "Aggregation_Operation":
    {
    "type":  ["Science Analysis Operations->Informations search->XLSX
  Processing of ...", "Element-wise->Math->Addition", "Science
  Analysis Operations->Batch Data Processing->", ...]
    }
  }
  ```
```

## B.5 PROMPT OF INTERGRATED DATA QUALITY CHECKING TOOL

---

**PROMPT OF EFFICIENT QUESTION CHECKING TOOL**

```
{Composition Reasoning Operations Prompt}
Question Checking

    □ Self-Containment:  The extent to which the question is fully
      specified and comprehensible without requiring additional
      external context.

    □ Retrieval Necessity:  The degree to which answering the
      question necessitates consulting external sources, while
      avoiding excessive disclosure of information within the
      question itself.

    □ Aggregation Necessity:  The question must include at least
      three different aggregation operations, ensuring that the
      answer cannot be obtained through direct retrieval.

    □ Clarity:  The precision and unambiguity of the cues or
      references embedded in the question that facilitate accurate
      data retrieval.  The clues will not lead to multiple feasible
      answers.

    □ Temporal Stability:  The property that the correct answer
      to the question remains consistent over time, unaffected by
      temporal changes (e.g., "Who was the immediate past president
      of the United States?").

    ---------------
Based on the above criteria, analyze the following data:
Question:  {}
Answer:  {}
Evidence_URLs:  {}
Please verify whether each item meets the standards and provide
advice for improvements.
```

---

## C DATA CONTAINMNATION ANALYSIS

We investigate potential data contamination between WebAggregatorQA and GAIA through two similarity analyses, with the results presented in Figure 11. First, to establish a baseline, we prompt an LLM to paraphrase the GAIA dataset, creating a new version termed GAIA-para. We then measure the similarity between the original and paraphrased GAIA questions. Second, we assess the cross-dataset overlap by sampling 100,000 pairs from the Cartesian product of the WebAggregatorQA training set and GAIA and computing their similarity. For all comparisons, we employ three metrics: a semantic similarity score calculated by Qwen3-Embedding-8B, ROUGE-L, and Jaccard similarity.

Figure 11 reveals a stark contrast: while GAIA shows significant similarity with its paraphrased counterpart, its overlap with WebAggregatorQA is negligible. This provides strong evidence that our WebAggregatorQA dataset is free from GAIA data contamination.

## D CLOSE SCREENING AGENT TRAJECTORIES

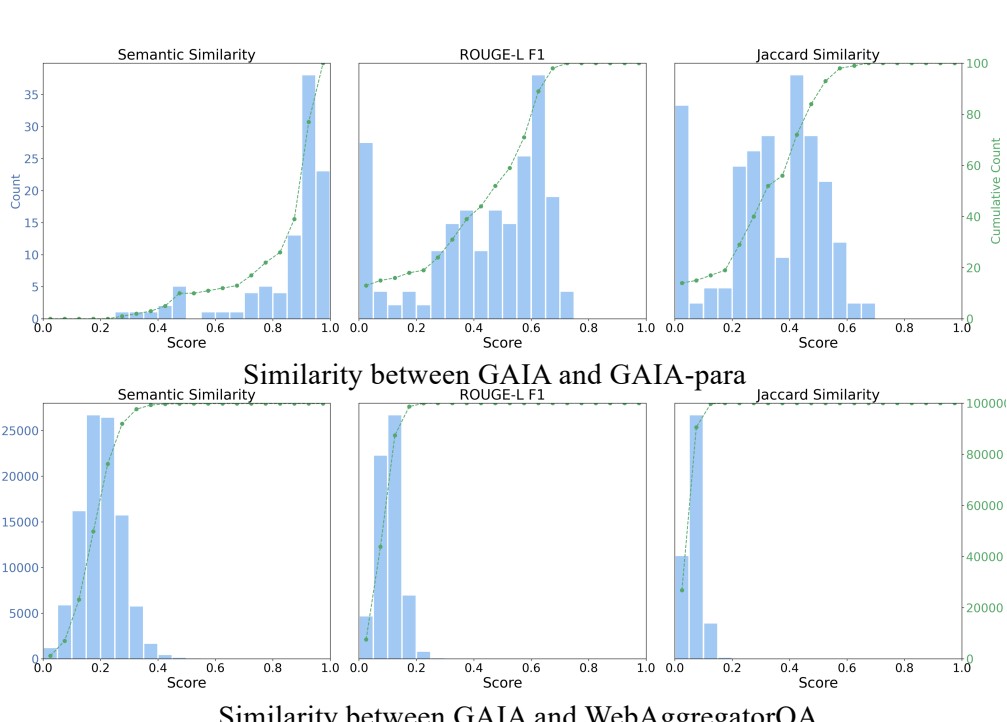

Figure 11: Data Contamination Experiment

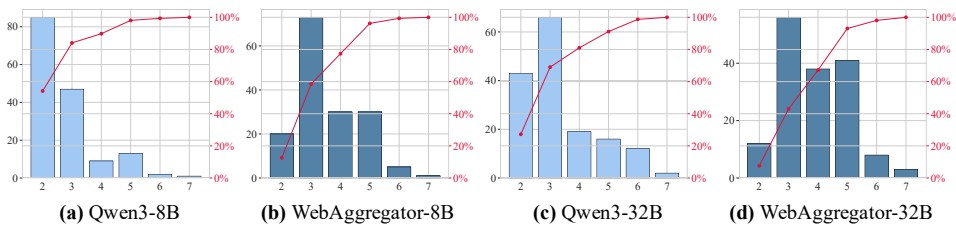

(a) Qwen3-8B    (b) WebAggregator-8B    (c) Qwen3-32B    (d) WebAggregator-32B

Figure 12: The distribution of unique tools used per trajectory, comparing the model before and after fine-tuning. After fine-tuning, WebAggregator utilizes a more diverse and balanced set of tools within each trajectory. In trajectories that involve only two tools, these are consistently `Search` and `Visit`.

## D.1 EXAMPLE 1

---

**Example 1**

> **User**
>
> Between 2014 and 2024, which year has the most new Trustees of the current Embry-Riddle Board of Trustees? And how many of the current Trustees received the Living Legend of Aviation?

> **Agent**
>
> First, I need to find the official Embry-Riddle Aeronautical University Board of Trustees page. This will likely contain the current list of Trustees and possibly historical data or links to archives. I will search for the Board of Trustees page and review its contents for the necessary information.
>
> ```
> 1 results = web_search(query="Embry-Riddle Aeronautical
>       University Board of Trustees")
> 2
> 3 print(results)
> ```

> **User**
>
> 1. [Embry-Riddle Aeronautical University: Board of Trustees](https://trustees.erau.edu/)
> 2. [Meet the Trustees](https://trustees.erau.edu/trustees)
> 3. [Trustee Ken Dufour and Chairman Mori ...](https://news.erau.edu/headlines/trustee-ken-dufour-and-chairman-mori-hosseini-mark-milestones-on-embry-riddles-board)
> 4. ...

> **Agent**
>
> The top search result is the official Embry-Riddle Board of Trustees page: https://trustees.erau.edu/.
> This page is likely to have a current list of Trustees and possibly links to historical data or news about new appointments.
> I will visit this page and review its contents to gather the information needed, especially looking for sections about current Trustees, their appointment years, and any links to archives or news about new appointments.
>
> ```
> 1 board_page = visit_webpage(url="https://trustees.erau.edu/")
> 2
> 3 print(board_page)
> ```

> **User**
>
> ...

**Agent**

Since there is no unified list or page displaying the year each trustee became a trustee, I will open each trustee's profile one by one to check and confirm whether they have received the Living Legend of Aviation award.

```
1  # Visit the papes of the first ten trustees
2  urls=[
3  "https://trustees.erau.edu/trustees/michael-amalfitano",
4  "https://trustees.erau.edu/trustees/michael-amalfitano",
5  "https://trustees.erau.edu/trustees/leanne-caret",
6  ...
7  ]
8  print(len(urls)) # There are {20} pages
9  for url in urls[:10]:
10     print(visit_webpage(url))
```

**User**

...
Mori Hosseini is the Chairman and CEO of ICI Homes, one of Florida's largest residential homebuilder/developers. Born in Iran, Mr. Hosseini was educated in London, England, ...
- Mori Hosseini (HonDoc '13; '78, '79, '82, DB)
- Member since **1999**
- Chair, Board of Trustees since 2014
- Chair, Committee on Trustees
...

**Agent**

I will continue to access other pages and record the years they became members.

```
1  trustees=[
2  "Michael Amalfitano" :{
3   "member year": "2021",
4   "living legend": False
5  },
6
7  "John Amore" :{
8   "member year": "2010",
9   "living legend": False
10 },
11 ...
12 "Mori Hosseini" :{
13  "member year": "2014",
14  "living legend": False
15 },
16 ...
17 ]
18 for url in urls[10:]:
19     print(visit_webpage(url))
```

> **Error Analysis:  Faulty Aggregation**
>
> Key information extraction error: 2014 v.s. 1999

**User**

...

**Agent**

```
1 final_answer=(\{"year":2014, "number of awardees": 3 \})
```

> **Error Analysis:  Faulty Aggregation**
>
> Faulty Aggregation while *Set->Filter->Counting*

**User**

END» answer: 2014, 3 # true answer: 2014&2018, 2

## D.2 EXAMPLE 2

─── **Example 2** ───

**User**

After one game's update introducing the Embrion and Old Bird, calculate the compound monthly CAGR of its average players from the update's release month until (but excluding) the next major patch's month, using steambase.io data. For the same period, calculate the CAGR of another game, which is a 4 player online co-op psychological horror game released in 2020. Present both game name and number as 'game: ±XY.ZT%'.

**Agent**

Step 1: Identify the game that received an update introducing "Embrion" and "Old Bird". I will use a web search to find which game this is and get the update's release month. After that, I will look for the next major patch's release month for that game.

```
1 search_query = 'game update introducing Embrion and Old Bird'
2
3 results = web_search(query=search_query)
4
5 print(results)
```

**User**

...

**Agent**

The update introducing Embrion and Old Bird is Version 50 of Lethal Company, released on April 13, 2024. I now need to find the release month of the next major patch after Version 50. I will search for "Lethal Company next major patch after Version 50" to determine this.

```
1 next_patch_search = web_search(query="Lethal Company next
    major patch after Version 50")
2
3 print(next_patch_search)
```

**User**

...

**Agent**

The most relevant sources suggest that after Version 50, the next major patch is speculated to be Version 70, but there is no explicit mention of an official release date. Since the question specifies to use steambase.io data, I should check steambase.io for the version history or patch notes for Lethal Company to see if there is a clear next major patch after Version 50. If not, I might have to use the next version listed or clarify if there is a clear, next major patch. I will visit the steambase.io Lethal Company page and look for update/version history.

```
1 lethal_company_steambase = web_search(query="Lethal Company
    steambase.io")
2
3 print(lethal_company_steambase)
```

> **Error Analysis: Faulty Aggregation**
>
> The next major patch is v60, on August 17, 2024. But the agent takes a guess.

**User**

...

**Agent**

The search did not provide a direct link to the Lethal Company page on steambase.io, but it did mention steambase.io as a source for player counts. I should visit steambase.io and look for the Lethal Company page, or try to directly access the relevant URL. Once there, I can look for monthly average player counts for the period starting April 2024 (the release month of Version 50) and check for any next major patch (likely Version 70) or any other official patch that is listed as "major" after Version 50.

```
1 lethal_company_url = "https://steambase.io/games/lethal-
      company"
2
3 page_info = visit_webpage(url=lethal_company_url)
4
5 print(page_info)
```

> **Error Analysis:   Underperformed Iterative Refinement**
>
> The next major patch is 'v60', on August 17, 2024. But the agent no longer checks whether 'v70' is reasonable and proceeds with subsequent steps.

**Agent**

The update introducing Embrion and Old Bird (Version 50) was released in April 2024. The next major patch after Version 50 is not explicitly listed, but the next major patch after 50 is Version 70, which is referenced in previous search results as the next major update.

...

Now, I will extract the average player counts for Lethal Company from April 2024 to March 2025 (12 months, not including April 2025), as that's the period from the update's release month until (but not including) the next major patch's month.

```
1 # Monthly average players for Lethal Company from steambase.io
      (April 2024 - March 2025)
2    lethal_company_averages = {    "2024-04": 24020,    "
   2024-05": 13345,    "2024-06": 12073,    "2024-07": 14299,
3
4    "2024-08": 16523,    "2024-09": 16036,    "2024-10":
   13545,    "2024-11": 14299,
5
6    "2024-12": 13120,    "2025-01": 17462,    "2025-02":
   15399,    "2025-03": 8886}
7
8 print(lethal_company_averages)
```

> **Error Analysis:   Underperformed Iterative Refinement**
>
> The next major patch is 'v60', on August 17, 2024. But the agent no longer checks whether 'v70' is reasonable and proceeds with subsequent steps.

**User**

The rest of the trajectory is omitted for readability.

## D.3 EXAMPLE 3

---

**User**

Using data from the World Bank for 2014-2023, calculate annual averages for: (1) GDP growth rate, (2) Inflation rate (consumer prices), (3) CO2 emissions per capita growth (excluding LULUCF), (4) Internet users (% population), (5) Women in parliament (%). Find the intersection between: (a) top 200 countries in (1),(3),(4),(5) and (b) bottom 200 countries in (2). Rank the countries in ascending order based on their CO2 emissions growth rate, with CO2 emissions growth rate rounded to 5 decimals. For the growth rate conditions mentioned above, if the World Bank provides growth rate data, prioritize using the World Bank's growth rate data. Otherwise, calculate it using the following formula: Growth Rate (%) = ( (data[n] - data[n-1]) / (data[n-1] + 0.00001) ) × 100 The small constant 0.00001 is added to avoid division by zero in cases where data[n-1] = 0. Give me the answer as 'Country: CO2 emissions growth rate'

---

**Agent**

This is a comprehensive data-rich task involving 5 World Bank indicators for all countries from 2014-2023. I need to:
1. Identify the exact World Bank indicator codes for each metric.
2. Download the datasets for each indicator (preferably from the World Bank API or CSV download links).
3. For each country and indicator, compute the annual averages for the period 2014-2023.
4. For indicators (1), (3), and (2): if WB has growth rate data, use it; if not, compute it using the formula provided.
5. Select the top 200 countries in (1) GDP growth, (3) CO2 per capita growth, (4) Internet users, (5) Women in parliament, and bottom 200 in (2) inflation. Find the intersection countries among top 200 in 1/3/4/5 and bottom 200 in 2.
6. Rank those by ascending order of their CO2 emissions growth rate, rounded to 5 decimals.

This is a multi-step data task. First, I will search for the official World Bank indicator codes for each variable.

```
queries = [

"World Bank indicator for GDP growth rate",

"World Bank indicator for inflation rate (consumer prices)",

"World Bank indicator for CO2 emissions per capita growth (
    excluding LULUCF)",

"World Bank indicator for Internet users percent the
    population",

"World Bank indicator for women in parliament percent"

]

for q in queries:
    print(f"{q}:", web_search(query=q))
```

---

**User**

...

**Agent**

I'll process the data **directly in memory**, reading from the zipfile into a 'csv.reader' and printing the header/first lines for inspection. approach requires no file system operations and should work given the allowed modules.

```python
def average(values):
    nums = [float(v) for v in values if v not in ('', None)]
    return sum(nums)/len(nums) if nums else None

def CO2_growth_rate(vals):
    # compute per-year growth rates, then average them
    result = []
    # Need at least two years to compute one growth rate
    vals_num = [float(v) if v not in ('', None) else None for v in vals]
    for i in range(1, len(vals_num)):
        if vals_num[i] is not None and vals_num[i-1] is not None:
            prev = vals_num[i-1]
            curr = vals_num[i]
            # formula with epsilon to avoid division by zero
            rate = ((curr - prev) / (prev + 0.00001)) * 100
            result.append(rate)
    return sum(result)/len(result) if result else None

indicator_avgs = {}
for key in indicator_data:
    indicator_avgs[key] = {}
    for code, vals in indicator_data[key].items():
        if key == "CO2_pc":
            avg = CO2_growth_rate(vals)
        else:
            avg = average(vals)
        indicator_avgs[key][code] = avg

# Print samples for all indicators to verify
for k in indicator_avgs:
    print(f"Averages for {k}:", list(indicator_avgs[k].items()
    )[:3])
```

**User**

...

**Agent**

I will obtain the countries with the highest CO2 emissions growth rates. Then I will begin calculating other metrics and finding their intersections.

```python
1  # Prepare (country, rounded growth rate) pairs for
       intersection countries
2  results = []
3  for code in intersection:
4      val = indicator_avgs['CO2_pc'][code]
5      if val is not None:
6          results.append((code, round(val, 5)))
7
8  # Sort by CO2 emissions growth rate, ascending
9  results_sorted = sorted(results, key=lambda x: x[1])
10
11 # Obtain the top 200 countries and growth rate
12 country_rate=results_sorted[:200]
13 print(country_rate)
```

**Error Abalysis: Error Accumulation in Aggregation**

Due to **the pre-reduction of the contry list in this step**, the final intersection differed from the actual results, causing **this trajectory to ultimately return an empty list**.

