# OpenReview forum: "WebAggregator: Scaling Complex Logical Information Aggregation for Web Agents Foundation Models"
_ICLR.cc/2026/Conference — ICLR 2026 Conference Withdrawn Submission_

### Official Review · Reviewer_U1EB · 2025-10-25

**Soundness:** 3
**Presentation:** 2
**Contribution:** 3
**Rating:** 4
**Confidence:** 4

**Summary:**

This paper, WebAggregator: Scaling Complex Logical Information Aggregation for Web Agent Foundation Models, presents a new paradigm for developing web agents capable of information aggregation rather than mere retrieval. The authors introduce WebAggregatorQA, a large-scale dataset automatically constructed through “Proactive Online Web Exploring” and “Complex Aggregation Logic Injection,” covering 10K samples from 50K websites across 12 domains. The dataset captures 12 categories of aggregation logic (e.g., mathematical, statistical, temporal reasoning) grounded in real web interactions. On top of this dataset, the authors train WebAggregator models (8B, 32B) based on the SmolAgents framework, achieving performance on par with or exceeding GPT-4.1 on GAIA-text and new benchmarks. The results reveal that even strong commercial models like GPT-4.1 and Claude 3.7 Sonnet underperform on the human-verified WebAggregatorQA test set, emphasizing the challenge of complex web information aggregation

**Strengths:**

The work tackles a critical bottleneck in web agent research—the lack of robust aggregation capabilities beyond information retrieval. The proposed data construction pipeline is scalable, verifiable, and largely autonomous, enabling reproducible and diverse task generation grounded in real-world web environments. The taxonomy of 12 logical operations provides fine-grained control over task complexity, bridging low-level data fusion with high-level reasoning. Empirical evaluations are rigorous and comprehensive, spanning multiple datasets (GAIA, WebWalkerQA, XBench) and baselines, demonstrating significant gains over prior open-source agents. The paper also shows strong generalization and efficiency, with smaller models (8B) achieving competitive results using only a fraction of the data. Overall, this is a technically solid and methodologically well-executed contribution toward building web agent foundation models.

**Weaknesses:**

Despite its innovation, the paper exhibits several limitations. The model diversity is insufficient—only Qwen-based models are explored, excluding more varied architectures or scales (e.g., Llama, Mistral, Gemini, Claude Opus). This limits generality and may bias the conclusions toward one training paradigm. The evaluation primarily relies on synthetic QA formulations, and the absence of end-to-end deployment testing under real dynamic web conditions weakens ecological validity. Furthermore, while the dataset captures aggregation logic, it does not explicitly disentangle reasoning from retrieval—leaving unclear whether improvements stem from aggregation training or memorization of structured tasks. The paper also lacks deeper error analysis or ablation on aggregation subtypes (e.g., temporal vs. statistical), and training details (e.g., efficiency, scaling laws) could be elaborated. Lastly, though claims of surpassing GPT-4.1 are impressive, comparisons against newer closed models (e.g., GPT-5, Claude 3.7 Opus, Gemini 2.5 Pro) would strengthen the claims of frontier performance.

**Questions:**

1. How does the model performance vary across different aggregation logic types (e.g., temporal vs. statistical reasoning), and what are the main failure modes?

2. Could the authors expand the benchmark by including more model families and scales, especially closed-source models like Claude 3 Opus, Gemini 2.5 Pro, or GPT-5, to better assess generality?

3. How does WebAggregatorQA handle dynamic and multimodal web content (e.g., changing JavaScript-rendered pages, tables, or charts), and can it adapt over time?

4. Have the authors considered integrating mechanistic interpretability or trace-level reasoning evaluation to better understand how the model aggregates information?

5. Can this data-construction pipeline be extended to multimodal or multi-agent settings, enabling aggregation of text, vision, and structured data jointly?

---

> ### Author Response · Authors · 2025-11-21
> **Author Response (1/2)**
>
> We thank the reviewers for recognizing our contributions and for their insightful comments. We address all concerns below, and the corresponding revisions have been incorporated into the **revised paper** (see the updated PDF with **changes highlighted in blue**).
>
> ---
>
> ### **[W1: Training more models]**
>
> First of all, we chose the Qwen series of models because currently Qwen is the most widely used and capable model in the field of agents and tool calling. Previous work has also used the Qwen series of models, which facilitates our performance comparison.
>
> We conducted experiments on Qwen2.5 and Qwen3 with 7B, 8B, and 32B parameter sizes. We believe this is sufficient to prove the validity of our data.
>
> ---
>
> ### **[W2: Our Experiments DO Run on Real Web]**
>
> We would like to clarify here to avoid any misunderstanding. **Whether it's our data construction or all experiments evaluating agent performance, they are conducted in real network environments.**
>
> Whether it's GAIA or our constructed WebAggregatorQA, the agent (WebAggregator) needs to perform searches, content parsing, file downloads, and then integrate the information into the final answer under real dynamic web conditions.
>
> **Here, we present a video demonstration in the supplementary materials for your reference.**
>
> ---
>
> ### **[W3: Retrieval v.s. Reasoning]**
>
> Thank you for this insightful question. We address your concern in two ways: (1) we show that the performance gains are from improved reasoning/aggregation rather than merely from better retrieval, and (2) the improvements of the model come from capabilities evolution rather than memorizing the training set.
>
> We address both points below.
>
> Regarding point (1):
>
> **The aggregation operations are important and challenging for current models. But tuning on WebAggregatorQA facilitates the LLMs to exhibit advanced reasoning and aggregation abilities.**
>
> First, we establish that aggregation is the core bottleneck: **success in information seeking does not guarantee success in information aggregation.** Even powerful models fail at this step. For example, when Claude-3.7-sonnet successfully retrieved all necessary documents for a task, its final accuracy was still only **42%** (Table 4).
>
> Second, our fine-grained analysis (Section 4.2) reveals that training fosters a more sophisticated agent, marked by **more efficient, diverse tool use and universally stronger aggregation capabilities, with absolute improvements exceeding 15% for each type.**
>
> **Evolution of Tool Usage Behaviors**: We examined the model's reasoning and tool-use behavior. Our analysis reveals a compelling shift in strategy: **WebAggregator learns to solve problems with fewer tool-calling steps, but simultaneously exhibits greater diversity in its tool selection within a single trajectory.** This key finding suggests that our training methodology moves the model beyond brute-force tool use, fostering a more sophisticated and strategic planning ability.
>
> **Evolution of Aggregation Capabilities**: We dissected the performance gains across various logical aggregation types. The results confirm that **our training yields universal improvements across all categories**, validating the comprehensive nature of our dataset. Concurrently, our analysis reflects the current limitations of web agents: the most complex types, namely "Scientific Analysis" and "Set Operations," remain formidable challenges despite the improvements. This provides a nuanced picture of the model's progress and underscores the difficulty of these advanced reasoning tasks.
>
> **Successfully Retrieved but Wrongly Aggregated**: Our data directly addresses the agent's core weakness: aggregation, not retrieval. For instance, fine-tuning boosts accuracy from a mere 9.7% to 35.7% on tasks where information retrieval was already perfect (Table 4, revised paper), demonstrating **a targeted improvement in the model's ability to aggregate information**.
>
> Regarding point (2):
>
> First, each data point in WebAggregatorQA is based on a completely different starting URL. Second, during the data annotation process, each instance underwent extensive manual intervention, making it extremely different from the original data.
>
> Furthermore, we now present an analysis of the overlap between our training set and GAIA-text in **Appendix C**. **The results show that no data contamination exists.**
>
> ---
>
> ### **[W4&Q1: Error Analysis across Aggregation Types]**
>
> As we mentioned in the previous response, the results confirm that our training yields universal improvements across all categories, validating the comprehensive nature of our dataset.

---

> ### Author Response · Authors · 2025-11-21
> **Author Response (2/2)**
>
> ### **[W5&Q2: Evaluate more commercial models]**
>
> Thank you for the suggestion. Our paper already includes extensive benchmarks on several state-of-the-art models, including Claude 3.7 Sonnet and GPT-4.1. These results consistently demonstrate that even the most advanced models struggle to solve these tasks on both our WebAggregatorQA dataset and the GAIA benchmark when relying solely on their internal knowledge.
>
> **To further address your point, we have run additional experiments on GPT-5. The results, now included in the revised manuscript, reinforce our original conclusion.**
>
> ---
>
> ### **[Q3: How WebAggregatorQA handles dynamic and multimodal web content]**
>
> To better illustrate our process, we provide **a video in the supplementary material**, showing the construction procedure of a WebAggregatorQA sample: the agent constructed the question, answer, and solutions after browsing web pages.
>
> Our data construction pipeline involves an agent that proactively visits web pages with a browser. Guided by high-level aggregation logic, it gathers information to construct question-answering pairs.
>
> To address your specific question about handling complex web elements: user tasks often require agents to process diverse sources, from static text to dynamic content like JavaScript-rendered pages, tables, and charts. To meet this need, we equipped our agent with a suite of tools capable of incorporating all these information types.
>
> Critically, during data construction, the agent waits for each page to fully render in the browser before capturing its content. This ensures that dynamic elements generated by JavaScript, as well as tables, charts, and figures, are all present. The agent's tools then parse this fully rendered information, which serves as the foundation for the entire QA generation process.
>
> ---
>
> ### **[Q4: Trace-level Evaluation]**
>
> For the trace-level evaluation, we assume that your meaning is that interpreting or using subgoal-matching to analyze the trace, providing an intermediate reward signal of how many proportion of the sub-tasks, aggregation operations have been achieved.
>
> First, most DeepResearch Agent evaluation sets out there adopt the setting of ground-truth answer matching and thus use accuracy (or Pass@k) as the evaluation protocol, due to the great sparsity of gold information on the internet, and the complex nature of agent tasks. Leading to effectively benchmarking and evaluating the ability of a complex web agent,  the answer needs to be easily verifiable [(BrowseComp, Wei 2025)](https://arxiv.org/abs/2504.12516). The major test set we follow, GAIA, only provides a ground-truth answer, and thus, we can only use pass@1 for evaluation.
>
> Actually, the WebAggregatorQA naturally includes solutions that involve intermediate checkpointing subtasks and reference web sources, which could be easily decomposed and serve as a valuable testbed for trace-level evaluation.
>
> We exhibit the decomposed logic structure of a construction question here (copied from Figure 8, page 9).
>
> > Question: Among all countries with a land area over 2 million square kilometers and whose population increased by less than 20% between 2010 and 2023, which country had the highest ratio of GDP growth rate to population growth rate over that period? What is the standard deviation (rounded to the nearest whole number) of its annual per capita GDP (in USD) from 2010 to 2023?
> >
> >
> > **Aggregation Logic Decomposition：**
> >
> > 1. *Element → Retrieve* → A: countries area, B: countries population
> > 2. *Element → Math* → C: population increase rate
> > 3. *Set → Filter* → D: countries with A>2M km², E: countries with C<20%
> > 4. *Set → Sets Composition* → F: intersection(D, E)
> > 5. *Element → Retrieve* → G: GDP and GDP growth rate of F
> > 6. *Element → Math* → H: ratio of G to C
> > 7. *Set → Filter* → I: top-1 of H
> > 8. *Element → Retrieve* → J: per capita GDP of country I
> > 9. *Sci. Analy. → Statis.* → K: std of J of country I
> >
> > **Answer → I: China, K: 2604**
> >
>
> ---
>
> ### **[Q5: Our Framework fully supports multi-modal]**
>
> Yes, our data construction pipeline is designed with native support for multimodality. In fact, our WebAggregatorQA dataset is inherently multimodal: the training set contains 12% multimodal (MM) data, and the test set includes 30 tasks that require multimodal reasoning, with a concrete example provided in Figure 10. Furthermore, the framework is extensible by design. By integrating new toolsets, the code we provide in the supplementary materials can be used to generate training data for any other modality, including speech.

---

> ### Author Response · Authors · 2025-11-26
> **Eager for Your Feedback and Further Discussion**
>
> Dear Reviewer **U1EB**,
>
> Following up on our rebuttal, here is a brief summary of the key actions we took in response to your feedback:
>
> - **Expanded Evaluation:** We ran **new experiments on GPT-5** to further validate our findings across state-of-the-art models.
> - **Live Web Confirmation:** We added a **video demonstration** to the supplement to unequivocally show all experiments run on the live, dynamic web.
> - **Reasoning vs. Retrieval Proof:** We proved our data teaches **true aggregation skills**, boosting accuracy of Qwen3-32B from 9.7% to 35.7% even when retrieval was perfect. We also verified **no data contamination**.
> - **Robust Framework:** We clarified that our framework natively handles **dynamic and multimodal content**, making it suitable for diversified requests.
>
> We hope this concise summary addresses your core concerns and are eager for any further discussion.
>
> Thank you for your time.
>
> Sincerely,
>
> The Authors

---

### Official Review · Reviewer_3EbL · 2025-10-31

**Soundness:** 3
**Presentation:** 2
**Contribution:** 3
**Rating:** 6
**Confidence:** 3

**Summary:**

The paper proposed a disciplined method to curate diverse web research tasks, which incorporates proactive online exploration and complex aggregation logic injection, and collected 6K trajectories to fine-tune LLMs for web-research tasks.
8B LLM fine-tuned with the collected trajectories achieved performance on par with GPT-4.1, and 32B LLM fine-tuned with the collected trajectories achieved performance higher than GPT-4.1 on GAIA benchmark.

**Strengths:**

* This paper introduced disciplined method to collect dataset for training web-based research agents at scale.
* Authors also provided WebAggregatorQA, which is challenging evaluation tasks aimed at evaluating research agents.
* Authors demonstrated the quality of the training dataset by achieving performance superior to previous baselines in GAIA benchmark.
* Comparison with comprehensive baselines and detailed analysis.

**Weaknesses:**

[W1] While the authors emphasize "information aggregation" over the "information seeking" in the abstract, empirical results demonstrating current research agents' inability to aggregate information is not provided. It would be better if the paper provides 1) failure modes of baselines (e.g., circumstances that baselines successfully seek web pages related to answer but failed to aggregate them in to a correct answer),  and 2) whether the agent trained with WebAggregatorQA dataset mitigate this issue.

[W2] In section 2.2.2, explanation about aggregation logic injection is quite hard to clearly understand. I could understand the detailed process by looking at the prompts in the Appendix. It would be better if there is a figure showing the process of QA pair is being synthesized from the exploration trajectory.

[W3] (minor) In the current draft, abstract comprises 2 paragraphs and is quite lengthy. It would be better to merge them into a single paragraph and shorten the contents.

**Questions:**

[Q1]. How can we ensure that there is no QA samples that overlap with evaluation task in GAIA benchmark?

[Q2]. In Appendix B.3, isn't the presented prompt for aggregation logic injection?

[Q3]. Do the authors have any plan to opensource the collected training dataset and the model checkpoint?

---

> ### Author Response · Authors · 2025-11-21
> **Author Response (1/2)**
>
> We thank the reviewers for recognizing our contributions and for their insightful comments. We address all concerns below, and the corresponding revisions have been incorporated into the revised paper (see the updated PDF with changes highlighted in blue).
>
> ---
>
> ### **[W1: Analysis about Aggregation Abilities Evolving]**
>
> We thank the reviewer for this insightful comment. We would like to respectfully clarify that we provided this evidence in our original manuscript and have further strengthened it with new analyses in our revision.
>
> First, we provided quantitative evidence for this failure mode in **Section 4.1, Page 8,** and supplied the zero-shot performance of Qwen3-32B. As shown in Table 4, even a powerful baseline like Claude-3.7-sonnet, which successfully retrieved all necessary reference documents in 38 tasks, still achieved an accuracy of only 42%. This result directly highlights the critical gap: **success in information seeking does not guarantee success in information aggregation**. We found that fine-tuning boosts accuracy from a mere 9.7% to 35.7% on tasks where information retrieval was already perfect (Table 4, revised paper), demonstrating a targeted improvement in the model's ability to aggregate information.
>
> Second, we have expanded this in Section 4.3, page 9 of our revised paper, and provide examples. Our qualitative analysis identifies two primary failure modes:
>
> 1. **Information Aggregation Failure**: This is the most common failure type. It occurs when the agent applies flawed reasoning, such as false causality or unsupported assumptions, fails to leverage key information, and thus misinterprets evidence.
>
> 2. **Error Propagation**: A single, minor error in an early aggregation step can derail the entire subsequent process. Due to limitations in self-correction, the agent either fails to identify the initial mistake or, in attempting to fix it, excessively lengthens the context, which in turn degrades its reasoning performance.
>
> Moreover, we conducted a fine-grained analysis comparing the model before and after training, as shown in **Section 4.2**.
>
> **Evolution of Tool Usage Behaviors**: We examined the model's reasoning and tool-use behavior. Our analysis reveals a compelling shift in strategy: **WebAggregator learns to solve problems with fewer tool-calling steps, but simultaneously exhibits greater diversity in its tool selection within a single trajectory.** This key finding suggests that our training methodology moves the model beyond brute-force tool use, fostering a more sophisticated and strategic planning ability.
>
> **Evolution of Aggregation Capabilities**: We dissected the performance gains across various logical aggregation types. The results confirm that **our training yields universal improvements across all categories**, validating the comprehensive nature of our dataset. Concurrently, our analysis reflects the current limitations of web agents: the most complex types, namely "Scientific Analysis" and "Set Operations," remain formidable challenges despite the improvements. This provides a nuanced picture of the model's progress and underscores the difficulty of these advanced reasoning tasks.
>
> ---
>
> ### **[W2: Presentation about Complex Aggregation Logic Injection]**
>
> Thanks for your valuable advice! We will reorganize the logic to better exhibit the QA synthesis procedure.
>
> Here we also make a brief re-introduction for you, and we provide **a video in the supplementary material** for your reference.
>
> (1) The agent explored the web pages and obtained the knowledge.
>
> (2) The agent then **refers to the complex logic injection prompt (Appendix B.3) covering logic types in Table 5**, selects the specific logics, and combines the knowledge into questions and answers (example comparison in Figure 8, page 14).
>
> (3) After QA construction, a two-stage critique-and-refinement quality enhancement stage begins. We use the LLM and prompt at Appendix B.5, to quickly check the questions, to make sure the aggregation needs, information retrieval needs, and other basic properties.
>
> (4) At last, an agent will be prompted to visit all of the reference URLs to ensure alignment among questions, answers, and reference URLs, using the prompt in Appendix B.4. 11.72 % unsatisfied samples are filtered.

---

> > ### Comment · Reviewer_3EbL · 2025-11-26
> >
> > Thanks the authors for the detailed response. Most of my concerns are resolved, and I will keep my rating as 6.

---

> > > ### Author Response · Authors · 2025-11-26
> > > **Response to Review 3EbL**
> > >
> > > Dear Reviewer 3EbL,
> > >
> > > Thank you for your response, and we are happy to see that our response addresses your concerns!
> > >
> > > Thank you for your time and feedback!

---

> ### Author Response · Authors · 2025-11-21
> **Author Response (2/2)**
>
> ### **[Q1: Overlap between WebAggregatorQA and GAIA]**
>
> Thank you for raising this important question. **We have implemented the following measures and analytically verified that no data contamination exists between WebAggregatorQA and GAIA.**
>
> Due to the inherent capability of web agents to access the internet, there is a potential risk that data exposed to search engines could be retrieved by the agent. **We are fully aware of this concern and have implemented comprehensive safeguards to address it.**
>
> - During data construction
>     - We explicitly excluded URLs associated with certain datasets (e.g., the GAIA project) from our initial seed URLs for data collection.
>     - We implemented a blocking mechanism based on URL prefixes to prevent the model from accessing datasets hosted on platforms like Hugging Face.
> - During agent evaluation:
>     - The same blocking mechanism was applied during evaluation, as we observed that search engines could potentially retrieve benchmark-related webpages. This blocking mechanism ensures the model cannot access such content.
> - Overlap analysis:
>     - To further address this concern, we conducted an overlap test between the complete training set of WebAggregator and GAIA. As detailed in **Appendix C**, our data contamination analysis could not find evidence that there is overlap between our dataset and GAIA.
>
> We believe these measures effectively mitigate potential data contamination issues.
>
> ---
>
> ### **[Q2: About Appendix B.3]**
>
> Appendix B.3 contains two prompts: one for "proactive online exploring" and another for "aggregation logic injection". We will make the reference to this section clearer in the main text.
>
> ---
>
> ### **[Q3: Open Source Plan]**
>
> We are committed to open-sourcing our (1) full training dataset, (2) data curation codebase, (3) modified agent framework, and (4) model checkpoints, as we mentioned in the Reproducibility Statement.
>
> Currently, we have provided the codebase (2) and framework (3) in the supplementary materials, while the other components were withheld due to file size limits. Once our paper is accepted, we will add a link to the repository in the final copy.

---

### Official Review · Reviewer_biXt · 2025-10-31

**Soundness:** 3
**Presentation:** 2
**Contribution:** 2
**Rating:** 2
**Confidence:** 4

**Summary:**

This paper introduces WebAggregator, an agent-based framework and dataset designed to improve web agents’ information aggregation capabilities. The authors propose an automated pipeline that constructs question–answer pairs by performing “Proactive Online Web Exploring” and “Complex Aggregation Logic Injection” on real web environments, resulting in a new dataset called WebAggregatorQA (about 10K samples from 50K websites). Using this dataset, the authors fine-tune open-source LLMs (Qwen2.5/3-7B/32B) to develop WebAggregator models, which reportedly outperform GPT-4.1 and Claude-3.7-sonnet on GAIA-text and the newly created benchmark. The paper argues that this dataset and framework strengthen information aggregation, a currently underexplored ability in web agents.

**Strengths:**

1. The paper is clearly written and well-structured, with detailed methodology and visualized pipelines.
2. The dataset construction process is systematic, and the authors demonstrate strong engineering execution.

**Weaknesses:**

1. The contribution seems more system-integration–oriented than research-driven. Although the implementation is impressive, the framework does not introduce a new learning objective, reasoning algorithm, or theoretical insight. The claimed innovations (such as “Proactive Exploring” or “Aggregation Logic Injection”) could be described more precisely to highlight what differentiates them conceptually from previous pipelines.
2.The prompt design within the agent likely has a substantial influence on the reported results, yet this factor is not thoroughly analyzed. Since prompts often dominate model behavior, the absence of prompt ablations makes it difficult to determine whether the improvements stem from the proposed framework or from more effective prompting strategies.
3. The data generation pipeline appears largely self-contained, with both question synthesis and answer derivation relying on the same model-driven process. This raises concerns about self-reinforcement bias, where models generate and validate their own outputs without external verification or human grounding.
4. The description of the “Complex Aggregation Logic Injection” step is vague and lacks operational details. It seems to rely on ad hoc prompt mappings rather than a formalized or reproducible algorithmic process.
5. Since the web environment is inherently dynamic, the “Proactive Online Web Exploring” process may yield non-deterministic outputs across runs. The paper does not describe any environment control or caching strategy, which undermines reproducibility.
6. Because the model is trained and evaluated on data derived from the same generation procedure, there is a risk of target leakage or self-consistency bias. Improvements may reflect distributional alignment rather than genuine reasoning gains.

**Questions:**

weakness

---

> ### Author Response · Authors · 2025-11-21
> **Author Response (1/3)**
>
> We thank the reviewers for recognizing our contributions and for their insightful comments. We address all concerns below, and the corresponding revisions have been incorporated into the revised paper.
>
> ## [Overall Takeaways]
>
> **On Contribution**: Our work is a research-driven paradigm, not system integration. Its core innovation is a new methodology to generate training data that teaches complex information aggregation, addressing a significant gap in prior research.
>
> **On Prompt Influence:** The model's strong performance stems from high-quality data, not prompt engineering. We ensured a fair comparison by using identical, minimal prompts for all models during evaluation, and our robust validation pipeline controls for any randomness in generation prompts.
>
> **On Self-Reinforcement Bias:** We agree that self-reinforcement bias is a critical concern in synthetic data generation. Our pipeline explicitly prevents this through a crucial **external validation step**. A separate agent verifies each generated QA pair against the **live web URLs**, ensuring the data is grounded in external reality rather than the model's own internal knowledge. This breaks any potential self-reinforcement loop.
>
> **On "Complex Logic Injection"**: Complex Logic Injection provides question logic formulation guidance; the final data quality is further guaranteed by our comprehensive quality validation procedures. As shown in our figures (Fig. 4, 5, 8), it consistently generates questions with diverse and high compositional complexity.
>
> **On Reproducibility:** The work is fully reproducible. While the web is dynamic, our results are based on a final static dataset, which we will release along with all code and models to ensure the community can verify and build upon our findings.
>
> **On Target Leakage**: There is no target leakage, because our training and evaluation sets are strictly separated. The model is trained on synthetic data but evaluated on a manually annotated held-out set and standard OOD benchmarks (like GAIA), proving it has learned generalizable skills.
>
> ### **[W1: Why Proactive Exploration and Complex Logic Injection are Important]**
>
> We thank the reviewer for their feedback. We respectfully clarify that the system we built is the necessary foundation to address a core research question: **How can we generate training data that teaches agents the complex skill of information synthesis, moving beyond simple fact retrieval?**
>
> Our contributions, **Proactive Online Web Exploring** and **Complex Logic Injection**, are our answer. They form a novel conceptual framework for generating data that fosters deep reasoning by solving two fundamental challenges:
>
> 1. **The Knowledge Challenge: Sourcing Complex, Non-Memorized Information.** To train a true research agent, we need data with logical relationships that are not common knowledge or already memorized by the LLM.
> **Proactive Online Web Exploring.** Prior work often relies on static, pre-built knowledge graphs (e.g., from Wikipedia), which are expensive to build and limited in scope. We propose a paradigm shift to dynamic online exploration, where the agent navigates the live web to automatically collect diverse, up-to-date knowledge. This dramatically reduces the manual data preparation workload while providing virtually unlimited diversity and complexity.
> 2. **The Reasoning Challenge: Requiring True Aggregation.** A deep research agent must do more than retrieve facts; it must analyze and synthesize them to derive new conclusions.
> **Complex Logic Injection.** Previous datasets often contain problems that can be solved with simple retrieval, overlooking the need for aggregation (as shown in Figure 8). Our method directly targets this gap by providing high-level aggregation guidance (e.g., "Scientific Analysis"), which the agent instantiates into concrete, multi-step operations based on the knowledge it has gathered.
>
> For example:
>
> > **Question:** Among all countries with a land area over 2 million square kilometers and whose population increased by less than 20% between 2010 and 2023, which country had the highest ratio of GDP growth rate to population growth rate over that period? What is the standard deviation (rounded to the nearest whole number) of its annual per capita GDP (in USD) from 2010 to 2023?
> >
> >
> > **Aggregation Logic Decomposition：**
> >
> > 1. *Element → Retrieve* → A: countries area, B: countries population
> > 2. *Element → Math* → C: population increase rate
> > 3. *Set → Filter* → D: countries with A>2M km², E: countries with C<20%
> > 4. *Set → Sets Composition* → F: intersection(D, E)
> > 5. *Element → Retrieve* → G: GDP and GDP growth rate of F
> > 6. *Element → Math* → H: ratio of G to C
> > 7. *Set → Filter* → I: top-1 of H
> > 8. *Element → Retrieve* → J: per capita GDP of country I
> > 9. *Sci. Analy. → Statis.* → K: std of J of country I
> >
> > **Answer → I: China, K: 2604**
> >
>
> We have included a video in the supplementary materials showing our agent in action.

---

> ### Author Response · Authors · 2025-11-21
> **Author Response (2/3)**
>
> ### **[W2: Influence of Prompts]**
>
> We appreciate your concern regarding prompt sensitivity on the agent performance.
>
> Our methodology ensures that the prompt setup does not cause significant fluctuations or bias in agent performance, data construction, or the evaluation of results.
>
> We argue that the impressive performance of WebAggregator comes from our **effective and stable data construction pipeline**, and the resulting high-quality training resource, WebAggregatorQA, not from fine-tuned, task-specific prompts used during inference. We address your concern by analyzing the three types of prompts used in our work:
>
> 1. **Inference Prompts: Standard and Fair.** For all experiments, we used the standard, minimal system prompts from the SmolAgents framework, which only describe tool functions. This exact same prompt was used for all baselines (e.g., GPT-4.1, Claude), ensuring a fair comparison. To further prove our method is not prompt-dependent, we replicated our training on the CK-Pro framework, which uses a completely different prompt structure, and achieved consistently strong results.
>
> |  | **GAIA-text** |
> | --- | --- |
> | Qwen3-8B-0shot | 15.7 |
> | Qwen3-8B-tuned | 43.7 |
>
> 1. **Data Generation Prompts: Controlled and Validated.** While prompts are an integral part of our methodology for creating complex questions, we designed a rigorous **three-step validation pipeline** to decouple final data quality from any prompt-related stochasticity. This pipeline includes: (1) an immediate review of generated questions, (2) a secondary in-depth verification by another agent using the reference URLs(filtering 11% of samples), and (3) rejection sampling to select only high-quality data, filtering any data with a misalignment between the question and the answer. This ensures the final dataset's quality is robust and standardized, regardless of initial prompt variations.
> 2. **Evaluation Prompts: Standardized.** Our LLM-as-judge evaluation prompt was adopted directly from established prior work (e.g., WebThinker), ensuring our evaluation protocol is standard, reproducible, and unbiased.
>
> In summary, by using fair inference prompts, a robustly validated data generation process, and standard evaluation prompts, we are confident our results are driven by the quality of our dataset, not prompt engineering.
>
> In summary, by maintaining strict consistency in inference prompts, treating the data generation prompt as a component of our robust and validated pipeline, and using standard evaluation prompts, we are confident that our conclusions are sound and not a byproduct of prompt engineering.
>
> ---
>
> ### **[W3: Self-reinforcement bias is not an issue in Data Generation Pipeline]**
>
> Thank you for posing your questions. **We would like to emphasize that the concern of self-reinforcement bias is not an issue here, for our comprehensive quality checking procedure, and we DO have external and extensive verifications for data quality.**
>
> Our primary goal is to generate a challenging and solvable training dataset, including questions and their ground-truth answers. The utility of our data lies in its ability to teach the agent specific skills, which is validated as long as the QA pairs meet our quality constraints:
>
> **Step 1**: An immediate review of all generated QA pairs. Checking all the questions is well-organized, information-seeking, and information aggregation needs to be solved.
>
> **Step 2:** A secondary, in-depth verification by another agent using the reference URLs. Ensure the alignment among questions, answers, reference URLs, and solutions. 11% unsatisfied samples are filtered.
>
> **Step 3:** Rejection sampling during training to select only high-quality data. In this procedure, Rejection sampling would further filter any data with a misalignment between the question and the answer.
>
> **The above pipeline ensures that the final dataset is robust, regardless of initial prompt variations.**

---

> ### Author Response · Authors · 2025-11-21
> **Author Response (3/3)**
>
> ### **[W4: About Complex Aggregation Logic Injection]**
>
> In the Complex Aggregation Logic Injection step, we synthesize QA pairs by applying pre-designed logic templates to the knowledge gathered previously.
>
> Our experiments demonstrate that this synthesis process yields data with a rich and diverse set of aggregation logics:
>
> **Diversity of Logic:** Figure 4 shows a wide range of logic types (e.g., "average," "table," "processing"). Our generation is dynamic; for instance, a Correlate prompt can yield a question about "Pearson correlation" in one case and "Jaccard similarity" in another.
>
> **Compositional Complexity:** Figure 5(d) demonstrates that each question is logically complex, combining at least two—and often four to six—different aggregation logics.
>
> **Comparative Complexity:** Figure 7 confirms that our data's logical complexity and reasoning demands significantly exceed those of other datasets.
>
> To ensure this quality, our pipeline includes a dedicated quality control step (Figure 2). We annotate and analyze the logic of each QA pair, systematically filtering out overly simple questions and balancing the distribution of different logic types across the dataset.
>
> ---
>
> ### **[W5: Dynamic Web may influence reproducibility]**
>
> Thank you for your question regarding reproducibility. **We would like to emphasize that the dynamic web would not be a problem in reproducing our work.**
>
> We have broken down the process into two key components:
>
> **Reproducing the WebAggregatorQA Data Construction Pipeline**: The process of building the dataset is subject to the inherent dynamism of the internet. Due to network fluctuations and evolving web content, running our pipeline with the same seed URLs may not yield identical QA pairs in every run. However, we view this non-determinism not as a limitation, but as a beneficial feature. It naturally introduces greater diversity into the generated dataset, which can be valuable for training more robust models. Most importantly, **to ensure that any generated dataset meets a consistent quality bar**, we employ a rigorous three-step validation process. This pipeline—which includes checks for question validity, alignment between the question/answer/sources, and rejection sampling—acts as a quality gate. It guarantees that while the specific data instances may vary, the overall quality, structure, and complexity of the resulting dataset are stable and reproducible.
>
> **Reproducing the WebAggregator Model and Training Results**: The model training and evaluation part of our work is fully reproducible. To ensure this, we are committed to open-sourcing all of our constructed resources. We have already included the complete source code in the supplementary materials, and the full dataset will be released publicly to facilitate future research.
>
> ---
>
> ### **[W6: Clarification on Evaluation]**
>
> We clarify that the model is **not** trained and evaluated on data derived from the same generation, and emphasize that **we have taken rigorous measures to ensure a strict separation between our training and evaluation data**, guaranteeing a fair assessment of our model's generalization capabilities.
>
> The training and evaluation sets are fundamentally distinct in their generation and composition, preventing any data leakage.
>
> **Training Set (WebAggregatorQA-train)**: This set was generated synthetically using the methodology detailed in Section 2.
>
> **Evaluation Sets**: We use a combination of in-domain and out-of-distribution (OOD) test sets:
>
> **WebAggregatorQA-test**: This is our held-out test set, which was manually annotated to ensure quality and to create a clear separation from the synthetically generated training data.
>
> **GAIA-text and other benchmarks**: These are established, OOD benchmarks, ensuring that the model is tested on problems it has not seen during training. A data contamination test is shown in Appendix C.
>
> While we introduce WebAggregatorQA as a new evaluation set, the performance of both our model and the baselines is assessed across a comprehensive suite of benchmarks. For fairness and consistency, many of the baseline scores on these benchmarks were taken directly from their original publications.
>
> As shown in Tables 2 and 3, this includes GAIA, WebWalkerQA, and XBench.
>
> Across all of these benchmarks, our model, WebAggregator, consistently outperforms previous baselines. For example, WebAggregator-32B outperforms GPT-4.1 and is on par with claude-3.7-sonet and all of the previous strong baselines.
>
> In summary, our evaluation protocol, which combines a carefully curated held-out test set with multiple standard OOD benchmarks, robustly demonstrates that our model has learned generalizable skills for deep research, rather than simply overfitting to the training data distribution. The consistent high performance across various benchmarks is strong evidence of this generalization.

---

> ### Author Response · Authors · 2025-11-26
> **Eager for Your Feedback and Further Discussion**
>
> Dear Reviewer **biXt**,
>
> We are writing to follow up on our rebuttal and to see if our revisions have addressed your concerns. We wanted to briefly highlight the key actions and clarifications we took in response to your feedback.
>
> - **On Contribution**: Our work is a research-driven paradigm, not system integration. Its core innovation is a new methodology to generate training data that teaches complex information aggregation, addressing a significant gap in prior research.
> - **On Prompt Influence:** We have confirmed that the model's performance gains are **driven by the quality of our data, not prompt engineering.** The `WebAggregatorQA` dataset itself is what imparts the generalizable aggregation skill.
> - **On Self-Reinforcement Bias:** Our pipeline is designed to **explicitly prevent self-reinforcement bias.** This is enforced by a critical external validation step, where a separate agent grounds each QA pair against live web sources to ensure fidelity to external reality.
> - **On "Complex Logic Injection"**: This component serves as a **high-level but basic guide for question formulation.** The final data quality and complexity are then guaranteed by our rigorous, multi-stage validation pipeline, as shown in  Fig. 4, 5, 8.
> - **On Reproducibility:** Our work is **fully reproducible.** Our robust data construction method is **specifically designed to handle the dynamic nature of the web**, mitigating randomness and producing a **final, static dataset.** We will release this dataset, along with all code and models, to ensure the community can easily reproduce our experiments.
> - **On Target Leakage**: We use strictly separate datasets for training (our generated data) and evaluation (a held-out set and OOD benchmarks like GAIA).
>
> We are keen to hear if these revisions have addressed your concerns and are available for any further discussion.
>
> Thank you for your time and consideration.
>
> Sincerely,
>
> The Authors

---

### Official Review · Reviewer_o4Zd · 2025-11-01

**Soundness:** 2
**Presentation:** 3
**Contribution:** 2
**Rating:** 4
**Confidence:** 3

**Summary:**

The paper addresses the purported critical limitation that existing web agents neglect complex information aggregation, focusing excessively on simple retrieval. The authors introduce an automated, scalable methodology to synthesize WebAggregatorQA, a training resource grounded in real web environments using Proactive Online Web Exploring and Complex Aggregation Logic Injection (12+ logical operations). Models fine-tuned on this data, WebAggregator, demonstrate improved performance, matching or surpassing models like GPT-4.1 on both the GAIA-text subset and the newly constructed, specialized WebAggregatorQA test set.

**Strengths:**

The primary strength lies in identifying and explicitly targeting the under-explored necessity of information aggregation in deep research agents. The paper delivers a robust, scalable data construction paradigm that mandates complex aggregation logic and utilizes heterogeneous, dynamic web sources, including files. Furthermore, the resulting WebAggregatorQA test set is shown to be highly challenging; even state-of-the-art zero-shot agents like GPT-4.1 (25.8%) and Claude-3.7-sonnet (28.3%) struggle significantly, effectively setting a difficult new benchmark and validating the importance of this aggregation focus.

**Weaknesses:**

The discussion on solutions for failure cases was not convincing to me. While it is beneficial to evaluate the limitations of the model's capabilities, the fundamental problem with GUI Agent evaluation is the lack of a unified paradigm, which results in highly noisy outcomes. For instance, could we not use a GUI Agent specialized only in interacting with the browser to acquire specific information for environment interaction and data collection, and then train a separate model dedicated to planning and information aggregation/reasoning? Given such possibilities, the training paradigm and experiments presented in this paper might become less persuasive. This is not a flaw of this specific paper, but rather reflects the difficulty in defining truly valuable problems within the agent field itself. What I was hoping to see was a truly detailed discussion on failure cases, including how to solve these problems under various paradigms, but this is lacking. Consequently, the insights provided by a portion of the paper (the training section) are not substantial enough, and in my opinion, this paper is therefore not suitable for inclusion in ICLR.

**Questions:**

1. What is the true motivation for conducting the model training experiments? What do you intend to verify? Have you considered the possibility of other training paradigms?
2. What are your next steps or ideas for addressing the failure cases?
3. WebAggregator uses Pass@1 for evaluation. What are your thoughts on finer-grained evaluation methods? What would be the benefits? What problems would they introduce? Why did you initially choose not to include a finer-grained evaluation?
4. WebAggregator explicitly excluded tools like Screenshot and Scroll from its training trajectory sampling. What was the rationale behind this decision?

---

> ### Author Response · Authors · 2025-11-21
> **Author Response (1/3)**
>
> We thank the reviewers for recognizing our contributions and insightful comments. We address all concerns below, and the corresponding revisions have been incorporated into the **revised paper**.
>
> ---
>
> ### **[W1: The paper's Core Contribution]**
>
> We believe our work and the reviewer's high-level perspective are highly aligned.
>
> 1. **Our Core Contribution**
>
> We would like to clarify that our core contribution is **not a new agent architecture, but a foundational, paradigm-agnostic solution an under-explored capability in web agents: deep information aggregation.** We offer both the methodology to generate such data and the resulting WebAggregatorQA dataset, for both training and evaluation. While alternative agent designs are possible, the fundamental need for this aggregation skill is universal. Our work provides the first systematic tools to train and evaluate this core capability, making it a foundational contribution to the field.
>
> Our work addresses two critical limitations in current agent research: resources that focus only on simplistic retrieval rather than true information aggregation, and datasets that lack the diversity of real-world sources like PDFs and figures. The severity of this gap is highlighted by our findings on the WebAggregatorQA-test, where even powerful models like GPT-4.1 achieved only 33.3% accuracy despite having retrieved all necessary information. This demonstrates a clear failure in aggregation, not just retrieval.
>
> To overcome these challenges, we introduce two key innovations. First, **Proactive Online Web Exploring** moves beyond static knowledge bases to a dynamic paradigm where the agent actively explores the live web, gathering diverse and up-to-date knowledge. Second, **Complex Logic Injection** moves beyond constructing simple retrieval tasks to target a more profound capability: the requirements to aggregate disparate information and reason over it (an example below from Figure 8).
>
> The effectiveness of our approach is validated by strong empirical results. On the GAIA benchmark, our fine-tuned Qwen3-8B model outperforms GPT-4.1, and our 32B model achieves performance comparable to Claude 3.7 Sonnet, confirming that our method significantly enhances an agent's ability to perform complex, information-driven reasoning.
>
> > Question: Among all countries with a land area over 2 million square kilometers and whose population increased by less than 20% between 2010 and 2023, which country had the highest ratio of GDP growth rate to population growth rate over that period? What is the standard deviation (rounded to the nearest whole number) of its annual per capita GDP (in USD) from 2010 to 2023?
> >
> >
> > **Aggregation Logic Decomposition：**
> >
> > 1. *Element → Retrieve* → A: countries area, B: countries population
> > 2. *Element → Math* → C: population increase rate
> > 3. *Set → Filter* → D: countries with A>2M km², E: countries with C<20%
> > 4. *Set → Sets Composition* → F: intersection(D, E)
> > 5. *Element → Retrieve* → G: GDP and GDP growth rate of F
> > 6. *Element → Math* → H: ratio of G to C
> > 7. *Set → Filter* → I: top-1 of H
> > 8. *Element → Retrieve* → J: per capita GDP of country I
> > 9. *Sci. Analy. → Statis.* → K: std of J of country I
> >
> > **Answer → I: China, K: 2604**
> >
>
> 2. **Experiment on Data Generalizability across Agent Paradigm**:
>
> We found that **training on WebAggregatorQA also significantly improves the foundation models of other agent frameworks.**
>
> We demonstrate the generalizability of the WebAggregatorQA-train by reproducing the whole trajectory collection, rejection sampling, and SFT process on another open-source agent framework, Cognitive Kernel-Pro. When using Qwen-3-8B as the backbone LLM, the resulting performance is as follows:
>
> |  | **GAIA-text** |
> | --- | --- |
> | Qwen3-8B-0shot | 15.7 |
> | Qwen3-8B-tuned | 43.7 |
>
> This experiment clearly shows that training on WebAggregatorQA provides a generalizable reasoning capability that significantly boosts performance, even in a different agent architecture. The performance of the 8B model becomes on par with GPT-4.1, demonstrating that our data, not our specific training setup, is the key driver of improvement.
>
> Furthermore, regarding the suggestion of a separate aggregation module, we offer two insights:
>
> **Aggregation is Intertwined with Planning**: In complex research tasks, aggregation and reasoning are not just final steps but are required iteratively throughout the process to inform subsequent actions. As shown in Figure 8 (the example above), solving intermediate aggregation steps is crucial for the agent to proceed. Decoupling this tightly integrated process can be suboptimal.
>
> **End-to-End Can Be More Effective**: Training separate specialized models does not guarantee superior performance. Our results show that our end-to-end fine-tuned WebAggregator-8B model outperforms the CK-Pro-8B model from a multi-agent system, suggesting that a holistic approach can be more efficient and effective.

---

> ### Author Response · Authors · 2025-11-21
> **Author Response (2/3)**
>
> ### **[W2: The discussion on failure cases ]**
>
> We thank the reviewer for pushing for a more detailed discussion on failure cases. We have expanded this in Section 4.3 of our revised paper and provide examples. Our qualitative analysis identifies two primary failure modes:
>
> 1. **Information Aggregation Failure**: This is the most common failure type. It occurs when the agent applies flawed reasoning, such as false causality or unsupported assumptions, fails to leverage key information, and thus misinterprets evidence.
>
> 2. **Error Propagation**: A single, minor error in an early aggregation step can derail the entire subsequent process. Due to limitations in self-correction, the agent either fails to identify the initial mistake or, in attempting to fix it, excessively lengthens the context, which in turn degrades its reasoning performance.
>
> We have conducted model behavior comparison before and after tuning in Section 4.2. Please refer to the response to [W3&Q1] for more explanations.
>
> ---
>
> ### **[W3&Q1: The insights from the training]**
>
> Our training is primarily designed to enhance the model's ability to handle complex information aggregation requirements.
>
> While the strong benchmark results validate the effectiveness of our method in improving the agent’s overall capabilities, we conducted a more fine-grained analysis in Section 4.2 (revised paper) and supply results in Section 4.1 to provide direct evidence on the improvement of the agents’ aggregation abilities, which we detail below.
>
> **Evolution of Tool Usage Behaviors**: We examined the model's reasoning and tool-use behavior. Our analysis reveals a compelling shift in strategy: **WebAggregator learns to solve problems with fewer tool-calling steps, but simultaneously exhibits greater diversity in its tool selection within a single trajectory.** This key finding suggests that our training methodology moves the model beyond brute-force tool use, fostering a more sophisticated and strategic planning ability.
>
> **Evolution of Aggregation Capabilities**: We dissected the performance gains across various logical aggregation types. The results confirm that **our training yields universal improvements across all categories**, validating the comprehensive nature of our dataset. Concurrently, our analysis reflects the current limitations of web agents: the most complex types, namely "Scientific Analysis" and "Set Operations," remain formidable challenges despite the improvements. This provides a nuanced picture of the model's progress and underscores the difficulty of these advanced reasoning tasks.
>
> **Successfully Retrieved but Wrongly Aggregated**: Our data directly addresses the agent's core weakness: aggregation, not retrieval. For instance, fine-tuning boosts accuracy from a mere 9.7% to 35.7% on tasks where information retrieval was already perfect (Table 4, revised paper), demonstrating a targeted improvement in the model's ability to aggregate information.
>
> ---
>
> ### **[Q2: Next steps or ideas for addressing the failure cases]**
>
> As mentioned in our response to your comments on the weaknesses, our new failure case analysis highlights several key areas for future improvement. Potential next steps include:
>
> (1) **Strengthening reasoning and self-correction capabilities**: Improving the agent's core planning and its ability to effectively identify and recover from errors. [Scaling Test-time Compute for LLM Agents](https://arxiv.org/abs/2511.09030)
>
> (2) **Enhancing problem decomposition capabilities:** Given the complexity of deep research tasks, especially in WebAggregatorQA with its intricate retrieval demands and sophisticated aggregation structures (as shown in Figure 8 and the previous example), enabling the agent to break down problems into solvable sub-tasks is crucial for improving performance. [Solving a Million-Step LLM Task with Zero Errors](https://arxiv.org/abs/2511.09030

---

> ### Author Response · Authors · 2025-11-21
> **Author Response (3/3)**
>
> ### [Q3: **Finer-evaluation]**
>
> Thanks for asking about finer-grained evaluation. We kindly ask what it means specifically by "finer-grained evaluation", but we clarify our usage of Pass@1 as the evaluation protocol here:
>
> First, most DeepResearch Agent evaluation sets out there adopt the setting of ground-truth answer matching and thus use accuracy (or Pass@k) as the evaluation protocol, due to the great sparsity of gold information on the internet, and the complex nature of agent tasks. Leading to effectively benchmarking and evaluating the ability of a complex web agent,  the answer needs to be easily verifiable [(BrowseComp, Wei 2025)](https://arxiv.org/abs/2504.12516). The major test set we follow, GAIA, only provides a ground-truth answer, and thus, we can only use pass@1 for evaluation.
>
> Second, despite the existence of other types of DeepResearch evaluation pipelines, e.g., report writing, it's outside the scope of our paper as it's a completely different brand of research.
>
> As for finer-grained evaluation, we do have our own unique ideas of using subgoal-matching to provide an intermediate reward signal of how many proportion of the sub-tasks have been achieved.
>
> WebAggregatorQA naturally includes solutions that involve intermediate checkpointing subtasks and reference web sources, which could be easily decomposed and serve as a valuable testbed for trace-level evaluation.
>
> ---
>
> ### [Question 4: **About Tools]**
>
>  Our base model is text-only and cannot directly process raw image inputs, which is why we excluded vision-based tools that provide image inputs to the foundation model.
>
> However, our agent framework is equipped with an image caption tool that converts visual information into textual descriptions. The text-only LLM can then reason over these generated captions to solve tasks that require understanding visual content. During training, the rejection sampling would ensure the agent is not trained on tasks that it is theoretically incapable of solving correctly.

---

> ### Author Response · Authors · 2025-11-26
> **Eager for Your Feedback and Further Discussion**
>
> Dear Reviewer **o4Zd**,
>
> We are writing to follow up on our rebuttal and to see if our revisions have addressed your concerns. We wanted to briefly highlight the key actions and clarifications we took in response to your feedback.
>
> - **On Our Core Contribution:** Our core contribution is to boost the aggregation capabilities by construction high-quality training data. And we conducted a new experiment showing it boosts performance on a completely different agent framework, then concluded our method is **paradigm-agnostic**.
> - **On Training Insights:** We concluded our training improves **aggregation abilities** of agents, not just retrieval. Newly added experiment shows the agent learns more efficient tool use and aggregation from WebAggregatorQA.
> - **On Failure Analysis and Future Work:** We conducted failure analysis which identifies key issues and informs concrete future improvements.
> - **On Evaluation** We concluded our evaluation methods are calibrated with previous works, and WebAggregatorQA naturally includes reference solutions that could be testbed for future sub-tasks evaluation.
>
> We are keen to hear if these revisions have addressed your concerns and are available for any further discussion.
>
> Thank you for your time and consideration.
>
> Sincerely,
>
> The Authors

---

### Author Response · Authors · 2025-11-26
**Paper Revision Summary**

Dear reviewers and ACs,

Thank you for your constructive feedback. We have addressed all comments in our individual responses and have revised the manuscript accordingly. The revised sections are highlighted in blue.

Our revisions focus on a deeper analysis of how our `WebAggregatorQA` dataset and its underlying construction methodology impact key agent capabilities, particularly in information aggregation, tool usage. We believe these changes have significantly strengthened the paper's contributions and clarity.

1. **Added a comparative analysis of pre- and post-training capabilities.** This includes the evolution of the model's tool-use behavior (Section 4.2, Figures 5 & 12) and significant improvements in its aggregation capabilities (Section 4.1, Table 4; Section 4.2, Figure 7).
2. **Included a failure mode analysis on WebAggregatorQA.** This analysis identifies common failure patterns, especially aggregation failures and offers insights for future improvements (Section 4.3, Figures D.1 & D.2).
3. **Updated Table 2 to include the performance of GPT-5** for a more comprehensive comparison.
4. **Expanded Figure 8** to feature a logical decomposition of tasks from both prior work and our WebAggregatorQA dataset.
5. **Added a data contamination analysis** to ensure the integrity of our evaluation (Appendix C, Figure 12).
6. In the supplementary materials, we have included **a video demonstrating the data construction process** for `WebAggregatorQA`. It shows the agent generating a data sample that includes the final answer, reference URLs, and a complete reference solution.

We believe that these revisions could address your concerns and substantially improv the quality of our work. We appreciate your reconsideration.

Best，

The Authors

---

### Note · Authors · 2026-01-05

I have read and agree with the venue's withdrawal policy on behalf of myself and my co-authors.